



# Diagnosing modeling errors of global terrestrial water storage interannual variability

Hoontaek Lee[1,2], Martin Jung[1], Nuno Carvalhais[1,3,4], Tina Trautmann[1], Basil Kraft[1], Markus Reichstein[1,4], Matthias Forkel[2], and Sujan Koirala[1]

[1]Max Planck Institute for Biogeochemistry, Germany
[2]Technische Universität Dresden, Institute of Photogrammetry and Remote Sensing, Germany
[3]Departamento de Ciências e Engenharia do Ambiente, Faculdade de Ciências e Tecnologia, Universidade Nova de Lisboa, Caparica, Portugal
[4]ELLIS Unit Jena at Michael-Stifel-Center Jena for Data-driven and Simulation Science

**Correspondence:** Hoontaek Lee (hlee@bgc-jena.mpg.de)

**Abstract.** Terrestrial water storage (TWS) is an integrative hydrological state that is key for our understanding of the global water cycle. The TWS observation from GRACE missions has, therefore, been instrumental in calibration and validation of hydrological models and understanding the variations of the hydrological storages. The models, however, still show significant uncertainties in reproducing observed TWS variations, especially for the interannual variability (IAV) at the global scale. Here,

we diagnose the regions dominating the variance of globally integrated TWS IAV, and sources of the errors in two data-driven hydrological models that were calibrated against global TWS, snow water equivalent, evapotranspiration, and runoff data: 1) a parsimonious process-based hydrological model, the Strategies to INtegrate Data and BiogeochemicAl moDels (SINDBAD) framework, and 2) a machine learning-physically based hybrid hydrological model (H2M) that combines a dynamic neural network with a water balance concept.

While both models agree with GRACE that global TWS IAV is largely driven by the semi-arid regions of southern Africa, Indian subcontinent and northern Australia, and the humid regions of northern South America and Mekong River Basin, the models still show errors such as overestimation of the observed magnitude of TWS IAV at the global scale. Our analysis identifies modeling error hotspots of the global TWS IAV mostly in the tropical regions including Amazon, sub–Saharan regions, and Southeast Asia, indicating that the regions that dominate global TWS IAV are not necessarily the same as those that dom-

inate the error in global TWS IAV. Excluding those error hotspot regions in the global integration yields large improvements of simulated global TWS IAV, which implies that model improvements can focus on improving processes in these hotspot regions. Further analysis indicates that error hotspot regions are associated with lateral flow dynamics, including both sub-pixel moisture convergence and across pixel lateral river flow, or with interactions between surface processes and groundwater. The association of model deficiencies with land processes that delay the TWS variation could, in part, explain why the models

cannot represent the observed lagged response of TWS IAV to precipitation IAV in hotspot regions that manifest to errors in global TWS IAV. Our approach presents a general avenue to better diagnose model simulation errors for global data streams to guide efficient and focused model development for regions and processes that matter the most.



# 1 Introduction

Terrestrial water storage (TWS) encompasses all storages of water over the land surface and in the subsurface, including groundwater, soil moisture, vegetation water, surface water, snow, and ice. As a major component of the global hydrological cycle, it controls the energy and biogeochemical fluxes (Famiglietti, 2004; Pokhrel et al., 2021). Furthermore, TWS is linked to occurrences of droughts, floods, as well as food production and human security over land (Tapley et al., 2019). In terms of processes, TWS buffers the drainage of precipitation, and partly supports transpiration and vegetation photosynthesis during

dry season (Chapin et al., 2011; Guan et al., 2014; Madani et al., 2020; Miguez-Macho and Fan, 2021), which in turn affects the land-atmosphere feedbacks. Globally, TWS is associated with the rate or amount of carbon exchange between land and atmosphere (e.g., Jung et al., 2017; Humphrey et al., 2018; Luo and Keenan, 2022; Wang et al., 2022). The TWS is, thus, a key variable to understand the global energy, water and carbon cycles and their interactions with climate change.

As TWS controls several mechanisms and processes over land, the TWS observations from the Gravity Recovery and

Climate Experiment (GRACE) satellite missions, launched in March 2002, have been extremely valuable in understanding global hydrological processes and storages (e.g., Kim et al., 2009), and in evaluating and improving hydrological models (e.g., Lo et al., 2010; Schellekens et al., 2017; Zhang et al., 2017; Trautmann et al., 2018). For example, GRACE TWS was used to study the effect of global climate change on sea-level rise (e.g., Reager et al., 2016; Scanlon et al., 2018); estimate continental discharge using a data assimilation framework (e.g., Syed et al., 2009); predict floods (e.g., Reager and Famiglietti, 2009); and

to quantify anthropogenic influence in the global hydrological cycle such as groundwater depletion (e.g., Rodell et al., 2009; Felfelani et al., 2017; Meghwal et al., 2019; Hosseini-Moghari et al., 2020; Liu et al., 2021) and dam construction (e.g., Awange et al., 2019). In relation to hydrological model evaluations and improvements, GRACE observations have been used to estimate model parameters and evaluate model simulations at regional (e.g., Lo et al., 2010), continental (e.g., Trautmann et al., 2018), and global (e.g., Kraft et al., 2022; Trautmann et al., 2022) scales. Constrained by GRACE observations, the uncertainties in

model predictions are reduced, and the same model simulations have contributed to better understandings of the global TWS variations (e.g., Kraft et al., 2022; Trautmann et al., 2022). For instance, Trautmann et al. (2018) calibrated a process-based hydrological model with multiple observation products including GRACE TWS to show that snow dominates mean seasonal TWS variability while liquid water dominates the interannual TWS variability over northern mid-to-high latitudes.

Despite the advances in modeling TWS, the errors in TWS simulations are not fully understood and addressed. For example,

Zhang et al. (2017) evaluated TWS estimates in four different global hydrological models against GRACE and revealed that the model performance varied significantly across basins, even within the same climate zone. Scanlon et al. (2018) also reported that TWS trends by global hydrological models were underestimated or had the opposite sign over basins across the globe, compared to GRACE-derived TWS trends. Particularly, a notable mismatch in TWS between models and GRACE is at the interannual temporal scale, which is relatively less understood compared to the long-term trend which has been extensively

studied to quantify the impact of anthropogenic activities on the water cycle (e.g., Rodell et al., 2009; Scanlon et al., 2018;





Meghwal et al., 2019). Studies have reported that global hydrological models performed relatively worse in reproducing observed global TWS interannual variability (IAV) compared to other temporal scales (e.g., Zhang et al., 2017; Kraft et al., 2022). Jensen et al. (2020) showed that the longer term variability of TWS is dominant in GRACE compared to seasonal variations in Earth system models; and suggested that the models are less reliable in reproducing the timing of peaks of TWS IAV across

years compared to the magnitude and frequency.

It is, nonetheless, imperative to understand the sources of modeling errors in the global TWS IAV as they may lead to misunderstanding of Earth's hydrological change such as the trend of water availability (e.g., Jensen et al., 2019; Scanlon et al., 2018); extreme events such as droughts and floods (e.g., Chen et al., 2010; Humphrey et al., 2016); and even the observed global water-carbon interactions (e.g., Jung et al., 2017; Humphrey et al., 2018, 2021) and land-atmosphere-climate interactions (e.g.,

Jensen et al., 2019). Despite findings that the global TWS variability is dominated by the humid tropical regions (e.g., Syed et al., 2008, 2009; Humphrey et al., 2016), there are obstacles in understanding the global modeling errors and attributing them to regions or specific processes. The spatial pattern of errors largely varies among models and even varies with the climate zone for the same model (Zhang et al., 2017). In addition, hydrological mechanisms underlying the TWS IAV modeling error are relatively unclear due to differences in the modeling structure. As TWS variations emerge from complex water cycle

dynamics and interactions among storages and water-carbon cycle linkages, the model errors may be due to different inherent assumptions and processes. Therefore, understanding the spatio-temporal contribution of global TWS IAV and its errors is necessary before effective model improvements can be implemented.

In this study, we first present a quantitative and consistent analysis of global TWS IAV from GRACE observation and two different data-intensive modeling frameworks: 1) a parsimonious process-based hydrological model, the Strategies to INtegrate

Data and BiogeochemicAl moDels (SINDBAD) framework, and 2) a machine learning-physically based hybrid hydrological model (H2M) that combines a dynamic neural network with a water balance concept (see Sect. 2.2.2 for details). Both modeling frameworks are heavily rooted on using observations, and include GRACE observations in the model parameter estimation and evaluation. As such, under ideal conditions, the models provide the simulations that agree the most with observations, and the model errors, if any, could be attributed to either missing model processes or observational uncertainties. As we employ

a covariance matrix analysis (see Sect. 2.1.2), we not only evaluate the global IAV, but also identify the regions that are most relevant to the IAV of global TWS in GRACE observations and the two models.

After analyzing the global TWS IAV, we apply the same method to TWS IAV modeling errors to identify the regions that contribute the largest to the errors in global TWS IAV, and to demonstrate which local scale mismatches lead to the errors (see Sect. 2.1.3). Lastly, we characterize the error regions by evaluating their associations with hydrometeorological variables, with

an aim to identify the potential missing mechanisms and/or processes that may lead to the modeling errors (see Sect. 2.1.4). We specifically address the following questions:

1. Which regions contribute the most to the global TWS IAV in observation and data-driven model simulations?

2. Which regions contribute the most to the modeling error for global TWS IAV?

3. What are the potential hydrological processes associated with the model errors?





## 2 Methods and data

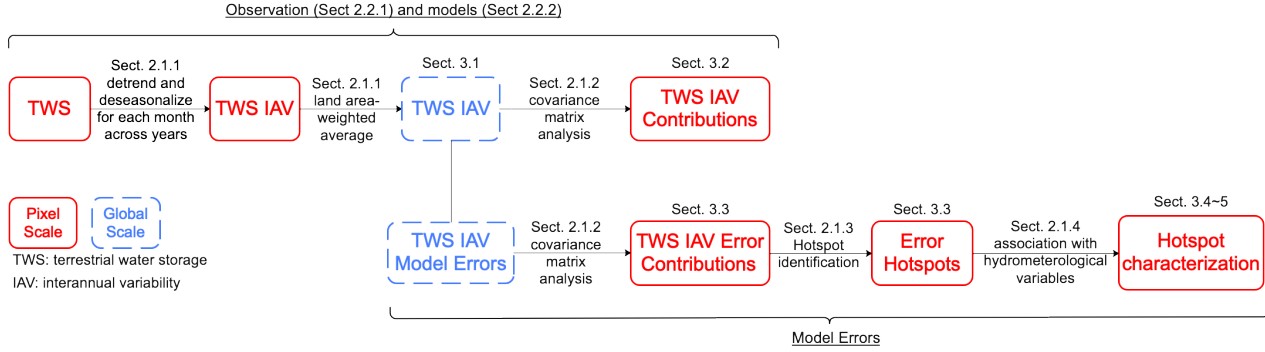

**Figure 1.** Schematic overview of the methodology to diagnose modeling errors in simulating the global TWS IAV. The sections of the manuscript that cover different aspects of the analysis are provided with Sect. headings.

To characterize the observed and modeled TWS variability and the associated modeling errors, we use GRACE TWS observations and two hydrological model simulations. An overview of the methodology and data used in this study is provided in Fig. 1. In this section, we present the details of the methods followed and data used.

### 2.1 Methods

### 2.1.1 Calculation of interannual variability

We calculated TWS IAV of each pixel as follows:

$$IAV_{i,m} = X_{i,m} - fit(X_{i,m}) \tag{1}$$

where $X$ is a time series of TWS anomalies after removing the pixel mean, $i$ is a pixel, $m$ is a month of the year, and $fit()$ is fitted values of a linear regression model over a time series of a month across years. In other words, the linear trend is fitted across all years for each month. The mean for the study period (April 2002–June 2017) was removed from the GRACE TWS anomalies or model TWS estimates for each pixel. This was done 1) to convert model TWS estimates to TWS anomaly estimates, and 2) to align different TWS anomaly products with the same baseline period.





Using the TWS IAV of each pixel, we calculated the globally integrated GRACE TWS IAV scaled by area of each pixel as follows:

$$IAV_{global,m} = \frac{\sum\limits_{i=1}^{n} (IAV_{i,m}) A_i}{\sum\limits_{i=1}^{n} A_i} \tag{2}$$

where $IAV_{global,m}$ is the global IAV for each month ($m$) which is calculated for both observation ($IAV_{global,m}^{obs}$) and model simulations ($IAV_{global,m}^{sim}$), $i$ is a pixel, and $A_i$ is area of pixel $i$. We multiplied TWS by pixel area so that the resulting TWS IAV is independent of the area of pixel which is dependent on map projection. Then, the modeled error is calculated as a residual of global TWS from observation and models as follows:

$$IAV_{global,m}^{err} = IAV_{global,m}^{sim} - IAV_{global,m}^{obs} \tag{3}$$

where $err$ indicates errors. A positive error therefore stands for an overestimation by a model.

### 2.1.2 Spatial attribution of the globally integrated signal

The variance of globally integrated TWS IAV and its error was attributed to each land pixel using a covariance matrix method. A covariance matrix calculated using the time series of land pixels contains the covariance of TWS IAV between a pair of pixels as the off-diagonal elements (for each combination of two different pixels) and the variance of TWS IAV in each pixel as the diagonal elements. By calculating the row (or column) sum of the covariance matrix scaled by the total sum, the contribution of a pixel to the variance of the globally integrated value can be quantified:

$$f_i = \frac{\sum\limits_{j=1}^{n} Cov(X_i, X_j)}{Var(\sum\limits_{i=1}^{n} X_i)} \tag{4}$$

where $f_i$ is the relative contribution of pixel $i$, $X_i$ is a random variable (i.e., TWS IAV or TWS IAV error) of the pixel $i$, $Var(X_i)$ is the variance of pixel $i$, and $Cov(X_i, X_j)$ is the off-diagonal element at the $i^{th}$ row and $j^{th}$ column in a $n \times n$ covariance matrix where $n$ is the number of land pixels in the globe. In the right hand side of the equation, the numerator is the sum of elements of $i^{th}$ row (or column); the denominator is the sum of all elements of the covariance matrix. This method quantifies the contribution of each pixel in a relative and normalized term; by definition, the sum of contributions of all pixels becomes one. It also considers the sign of contribution. A large positive covariance for a pixel means that the pixel increases the global variance, implying that the pixel has a similar temporal pattern as the global one. In contrast, a large negative covariance means that the pixel compensates with the global variability. The calculated $f_i$ is mathematically additive.

### 2.1.3 Hotspot identification

We calculated the $R^2$ between globally integrated time series of GRACE and models (Sect. 2.1.1) to evaluate model performance of TWS. The $R^2$ values were iteratively calculated after trimming out pixels that have large positive contributions to





the variance of global TWS IAV errors ($f_i$ in Eq. 4). The positive large covariances were trimmed because the purpose of this study is to identify pixels that contributed to the emerging global variability of TWS errors; in this method, pixels with positive contributions shape the global signal while pixels with negative contributions partly compensate for the signal. For the trimming percentile, 0.0 (i.e., no trimming), 0.1, 0.3, 0.5, 1, 2, 3, 5, 7% and from 10% to 50% with 5% intervals were used. An increase in $R^2$ after trimming suggests that the pixels of large positive contributions negatively affect the model performance in simulating TWS IAV. In other words, correcting simulations in those pixels can improve the model performance at the global scale.

### 2.1.4 Association with hydrometeorological variables

After the error hotspots are identified, we compare the time series of TWS and precipitation IAVs at the regional scale for error hotspots within selected SREX regions (Sect. 3.4; see Fig. B1 for the SREX regions) to diagnose TWS IAV errors. This association is simply quantified using the correlation coefficient between TWS and precipitation IAVs.

In addition, we associate the error with selected hydrological variables and we test if the variable can characterize the hotspots from non-hotspots. As both the models were forced by the climate variables, we assume that the model errors are less likely to be associated with climate characteristics and select variables that are related to missing model processes (see Sect. 2.2.3).

The association between the hotspots and the selected variables are inferred by comparing the histograms of hydrometeorological variables in hotspots and non-hotspots (see Sect. 2.1.3 for the hotspot identification) regions. The histograms, for hotspots and non-spots, of the different variables were computed across equal-size bins. We additionally provide probability density curve that was estimated using the Gaussian kernel whose bandwidth was determined with the Scott's rule of thumb. We then calculated the difference in the bin heights of the two histograms (hotspots minus non-hotspots) to find if the variable characterizes hotspots from non-hotspots. Positive (negative) differences mean that error hotspots have a higher (lower) probability density than non-hotspots for the given values of selected variable. Based on the differences, we assume that a variable which shows a clear positive difference in the histograms of hotspots and non-spots are influential in and potentially the source of error in the models.

### 2.2 Data

In this section, we introduce the datasets used in the analysis. First, we introduce the GRACE TWS observation data. This is followed by brief descriptions of two hydrological model simulations used to compare against GRACE. Lastly, we summarize the ancillary hydrometeorological data used for characterizing error hotspots.

### 2.2.1 GRACE TWS observation

For the TWS observation, we used a release 6 (RL06) mass conservation (mascon) solution Level-3 monthly TWS anomaly product provided by the Jet Propulsion Laboratory (JPL) (Wiese et al., 2016). The mascon solution is known to have lower





leakage errors compared to the traditional spherical harmonic solution (Scanlon et al., 2016). The product is provided at 0.5° × 0.5° spatial resolution, even though the native observational resolution of GRACE satellites is 3° (Wiese et al., 2016). The data is available for the period of April 2002 to June 2017. GRACE provides a vertically integrated estimate of TWS variations as anomalies relative to the observations from January 2004 to December 2009. A glacial isostatic adjustment has been applied

using the ICE6G-D model from Peltier et al. (2018) to exclude signals due to solid Earth deformation. The coastline resolution improvement filter and a set of scaling factors were applied to reduce leakage errors between lands and oceans (Wiese et al., 2016). Nevertheless, GRACE TWS still contains errors from measurement and leakage. Generally, GRACE errors 1) increase with decreasing latitudes toward the equator due to the polar orbit of the twin satellites (Frappart and Ramillien, 2018), and 2) are large at land-ocean boundaries due to the signal leakage (Wiese et al., 2016). The scaling factors are also incomplete in

ice-covered regions and near inland water bodies where the land surface model derived scaling factors are less reliable (Wiese et al., 2016).

### 2.2.2 Global hydrological model simulations

We analyzed simulations of TWS variations from two data-driven global hydrological modeling approaches: a) a conceptual hydrological process-based model configured using a model-data-integration framework, the strategies to integrate data and

biogeochemical models (SINDBAD), and b) the hybrid hydrological model (H2M) using a hybrid approach that fuses hydrological balance equations with a dynamic neural network. Both models were calibrated to hydrological data streams including TWS observations from GRACE. This makes the model errors less likely to be caused by parameter values, allowing us to focus more on the model structure and represented processes while analyzing errors. Owing to the use of observational data, SINDBAD shows a similar or better performance in simulating TWS seasonality and IAV compared to an ensemble of global

hydrological and land surface models (Trautmann et al., 2018). H2M has also been applied for different climatic regions over the globe to simulate TWS seasonality and interannual variability, where it is shown that H2M is capable to learn key patterns of the global water cycle components and has a comparable performance and better local adaptivity, compared to four state-of-the-art global hydrological models (Kraft et al., 2022). In summary, the two model simulations used here provide state-of-the-art where the model parameters and processes are partly learnt from GRACE observations, and are comparable to

state-of-the-art hydrological models commonly used.

SINDBAD, constrained with multiple observational data streams, has been successfully applied over northern mid- to high-latitudes (e.g., Trautmann et al., 2018) and over the globe (Trautmann et al., 2022) for simulating seasonality and interannual variability of TWS and its components. SINDBAD is a simple conceptual 4-pool water balance model with spatial variations of hydrological parameters depending on remote-sensing and statistical estimates of vegetation fraction and soil water capacity

(Trautmann et al., 2022). SINDBAD TWS consists of snow, soil moisture divided into shallow and deep components, and delayed moisture storages. Model parameters were constrained using TWS, snow water equivalent, evapotranspiration, and runoff observations, as well as vegetation characteristics including vegetative activity and maximum rooting depth.

The hybrid hydrological model (H2M, Kraft et al., 2020, 2022), on the other hand, uses a hybrid approach that fuses hydrological balance equations and a neural network, and provides an even stronger data-driven estimate of TWS variations.





In H2M, hydrological balance equations define three storages: snow water equivalent, soil water deficit (storage is represented as the additive inverse/opposite of the deficit), and groundwater. The model is driven by data of time-varying meteorological forcing and time-static soil and land characteristics. Five parameters (soil recharge fraction, groundwater recharge fraction, fast runoff fraction, snowmelt coefficient, and evaporative fraction) are estimated by a dynamic neural network and used as time-varying parameters in the balance equations. The first three parameters partition incoming moisture fluxes and the last

two quantify snow melt and evapotranspiration, respectively. The dynamic neural network updates each parameter value as a function of to the storage status of the previous time step, meteorological forcings of the present time step, and static variables for each pixel globally, resulting in spatiotemporally varying hydrological parameters that enables a large data adaptability of the hybrid approach. Observational constraints of TWS, snow water equivalent, evapotranspiration, and runoff were used for model tuning, and some static variables such as elevation, soil property, land cover type were used as additional model inputs.

While both modeling frameworks are predominantly data-driven, there are some key differences between them. First, H2M is more flexible and adaptive to the data used as H2M deploys the dynamic neural network to partition fluxes. Second, two models use additional data sets for the parameter estimation in addition to hydrological pools and fluxes that are used as constraints. SINDBAD uses vegetation-related input such as vegetation greenness index and rooting depth, although the vegetation variables do not vary spatially and interannually. H2M additionally uses some static variables related to soil properties, topog-

raphy, and land cover. Lastly, the structure of water pools is different. SINDBAD has two soil layers, an upper one with a fixed moisture capacity (4 mm) and a deeper one with a moisture capacity estimated from different datasets during the calibration process. H2M, on the other hand, approximates soil moisture as a cumulative deficit via a data-driven approach that does not require a prescription of the soil moisture storage capacity. As well, H2M implicitly learns the layering of soil moisture to a certain extent (Kraft et al., 2022). In SINDBAD, transpiration supply has access to the deeper soil moisture layer. H2M implic-

itly represents it as a function of soil water deficit. SINDBAD represents groundwater storage via deep and delayed storages, which are interacting through drainage flux and capillary rise, while H2M directly represents a single groundwater storage that is fed by a drainage flux without capillary rise. The delayed moisture storage of SINDBAD implicitly represents all water storage components that may delay the loss of moisture from the system and that includes surface water variations as well, while H2M does not account for surface water storage. In both models, a part of the excess throughfall forms fast direct runoff;

while baseflow (or slow runoff) is generated from delayed moisture storage in SINDBAD but groundwater in H2M. Finally, the TWS is a sum of snow, soil, deep and delayed storages for SINDBAD, and snow, soil and groundwater for H2M.

For consistency, a common set of forcings and constraints were used to force the models and calibrate the model parameters at 1° × 1° spatial and daily temporal resolutions (see Appendix A).

### 2.2.3 Ancillary data

The ancillary data were used to explore the potential association of modeled TWS errors with hydrometeorological variables (Table 1). All variables were aggregated into the 1° spatial and monthly temporal (for temporally-varying variables) resolutions using the land area of each pixel. In this section, we briefly summarize the dataset used.





**Table 1.** Data used to associate with modeled terrestrial water storage (TWS) errors.

| Variable | Product name | Original spatial resolution | Original temporal resolution | Reference |
|---|---|---|---|---|
| TWS | GRACE Tellus JPL RL06M v1 with CRI v1 | 0.5° | Monthly | Wiese et al. (2016) |
| | SINDBAD simulation | 1.0° | Daily | Trautmann et al. (2022) |
| | H2M simulation | 1.0° | Daily | Kraft et al. (2022) |
| Precipitation | GPCP 1dd v1.3 | 1.0° | Daily | Huffman et al. (2001) |
| | MSWEP V2.8 | 0.1° | 3-hourly | Beck et al. (2019) |
| Air temperature | CRUJRA v2.2 | 0.5° | Daily | Harris (2021) |
| Net radiation | CERES SYN1degEd4A | 1.0° | Daily | Wielicki et al. (1996) |
| Surface water occurrence and recurrence | GSWE | 30m | Static | Pekel et al. (2016) |
| River water storage | TRIP simulation with the input of runoff from SINDBAD | 1.0° | Daily | Oki and Sud (1999) (for TRIP model) |
| Wetlands fraction | | 15" | Static | Tootchi et al. (2019) |
| Transpiration water source | | 30" | Static | Miguez-Macho and Fan (2021) |

**Climate variables**

As the main drivers of TWS variabiality over land, we compared the IAV (i.e., Eq. 1) of the three forcing variables (precipita-
tion, air temperature, and net radiation) with TWS IAV (Sect. 3.4). We also calculated standard deviation of IAV of the forcing
variables of each pixel and compared the distribution of the standard deviation between error hotspots and non-hotspots to char-
acterize the error hoptosts. An additional observation-based precipitation product was used to check if the patterns vary across
different precipitation products. For that, we used the Multi-Source Weighted-Ensemble Precipitation, version 2.8 (MSWEP
V2.8, Beck et al., 2019) product that merges rain gauge, satellite, and reanalysis precipitation estimates.





### River water storage

The river water storage wRiver is considered as a proxy of surface water quantity. We assume that the regions with a large wRiver have a significant contribution of surface water component to the total TWS. For this analysis, wRiver was calculated using the Total Runoff Integrating Pathways (TRIP) river routing model (Oki and Sud, 1999) with the input of runoff from SINDBAD model simulations. Daily wRiver) was averaged to monthly values, and then monthly wRiver was converted to the absolute storage by multiplying the area of each pixel. The volumetric unit was used because the absolute storage, not the equivalent height, was used to calculate model biases (see Sect. 2.1.1). The maximum wRiver of a pixel ($\text{wRiver}_{\max}$) during the study period (April 2002–June 2017) was used for comparing its probability density distribution between error hotspots and non-hotspots. We applied log transformation to $\text{wRiver}_{\max}$ before comparing the probability density distribution to alleviate its skewed distribution.

### Wetlands fraction

To distinguish different surface water dynamics that are either associated with permanent river water bodies or seasonally dynamic storage interactions, we consider the wetland fractions dataset from Tootchi et al. (2019). The dataset contains seven composite maps that includes wetlands of two classes: 1) regularly flooded wetlands created from three satellite-derived datasets of open-water and inundation, and 2) groundwater-driven wetlands from water table depth estimations of Fan et al. (2013) and three topographic indices with two thresholds. Among the seven composite maps, Tootchi et al. (2019) provides two composite maps that showed the best similarity scores to reference datasets. The two composite maps have many similarities such as the spatial distribution and extent of wetlands (Tootchi et al., 2019); and the one using the topographic index (CW-TCI15) was used in this study. The dataset was averaged into 1° spatial resolution. The probability density distribution of each class of wetlands, excluding their intersection, were compared between error hotspots and non-hotspots. Regularly flooded wetlands represent the river storage dynamics, while groundwater-driven wetlands indicate the propensity for interactions among soil water, groundwater, and surface water due to shallow water table depth.

### Groundwater usage by vegetation

We evaluate the relevance of different moisture sources in the evaporative process, which in turn affects the TWS dynamics, by evaluating the associations of moisture sources for transpiration with occurrences of hotspots. For that, we used the transpiration source partitioning dataset by Miguez-Macho and Fan (2021). The dataset considers four sources of transpiration: 1) precipitation in the current month, 2) past precipitation stored in unsaturated soils, 3) locally recharged groundwater via capillary flow, and 4) remotely recharged groundwater from uplands to lowlands. Note that the remotely recharged groundwater indicates groundwater converged topographically (Miguez-Macho and Fan, 2021), which was originally estimated at a high spatial resolution of 30 arc-seconds. Using the map of annual mean contribution aggregated to 1° (one used for Fig. 2 in Miguez-Macho and Fan (2021)), we compared the probability density of contributions of capillary flow and converged groundwater from uplands between error hotspots and non-hotspots.





**Surface water existence**

To investigate the relationship between TWS IAV error and propensity of existence of surface water, we used a Landsat-based data for surface water existence from the global surface water explorer (GSWE, Pekel et al., 2016). Among the set of GSWE variables, we used two static variables: occurrence and recurrence. Briefly, the occurrence is calculated with the following steps: 1) calculate the ratio of the number of valid observations with surface water detected to the total number of valid observations for each month of the year (i.e., twelve values in total) and 2) average the twelve values. This two-step calculation normalizes the occurrence against the seasonal variation of the number of observations, which otherwise the occurrence is biased by the temporal variation of valid observations of the pixel (Pekel et al., 2016). Recurrence is calculated as the number of years with at least one valid observation with surface water detected divided by the number of years with at least one valid image collected. The occurrence thereby provides the general overview of the frequency of surface water existence, while recurrence represents the interannual fraction of the surface water.

For occurrence, to calculate the spatial fraction of surface water domination, we first assigned a value of 1 for the raster of 30 arc-seconds when the occurrence value is larger than a threshold value (50%) and 0 otherwise. We then spatially averaged the raster of 30 arc-seconds to $1°$ spatial resolution using valid observations only. For recurrence, we spatially averaged the raster of 30 arc-seconds to $1°$ spatial resolution as the data is already suitable for representing temporal fraction of surface water presence. A coastline mask (Sayre et al., 2019) was applied to occurrence and recurrence data sets before the aggregation to exclude possible biases of high occurrence and recurrence in coastal regions. We used the derived occurrence as a proxy of spatial existence. For the temporal existence of surface water, we used a product of occurrence and recurrence (i.e., recurrence multiplied by (1–occurrence). The product accounts for the interaction between occurrence and recurrence and excludes pixels dominated by permanent existence or a complete lack of surface water bodies (i.e., either recurrence or 1–occurrence becomes zero). This assigns larger values to the regions where presence of surface water is temporally erratic (high recurrence and low occurrence) but significant, which may lead to a large IAV of TWS.

## 3    Results

In this section, we first compare modeled and GRACE global TWS IAV to get a glimpse of the model performance and errors (Sect. 3.1). We continue to introduce pixel-wise contributions to the variance of global TWS IAV in GRACE and models (Sect. 3.2), and model errors (Sect. 3.3), followed by inspecting the time series of TWS IAV and precipitation IAV for selected regions (Sect. 3.4). We finally investigate the association of the global TWS IAV errors to hydrological variables, including surface water dynamics and groundwater usage by vegetation (Sect. 3.5).

### 3.1    Global TWS interannual variations

SINDBAD and H2M reasonably reproduce the observed global TWS IAV by GRACE ($R^2$ of 0.49 and 0.51 for SINDBAD and H2M, respectively) (Fig. 2). The modeled and observed variations are generally similar in the sign of variations (positive or





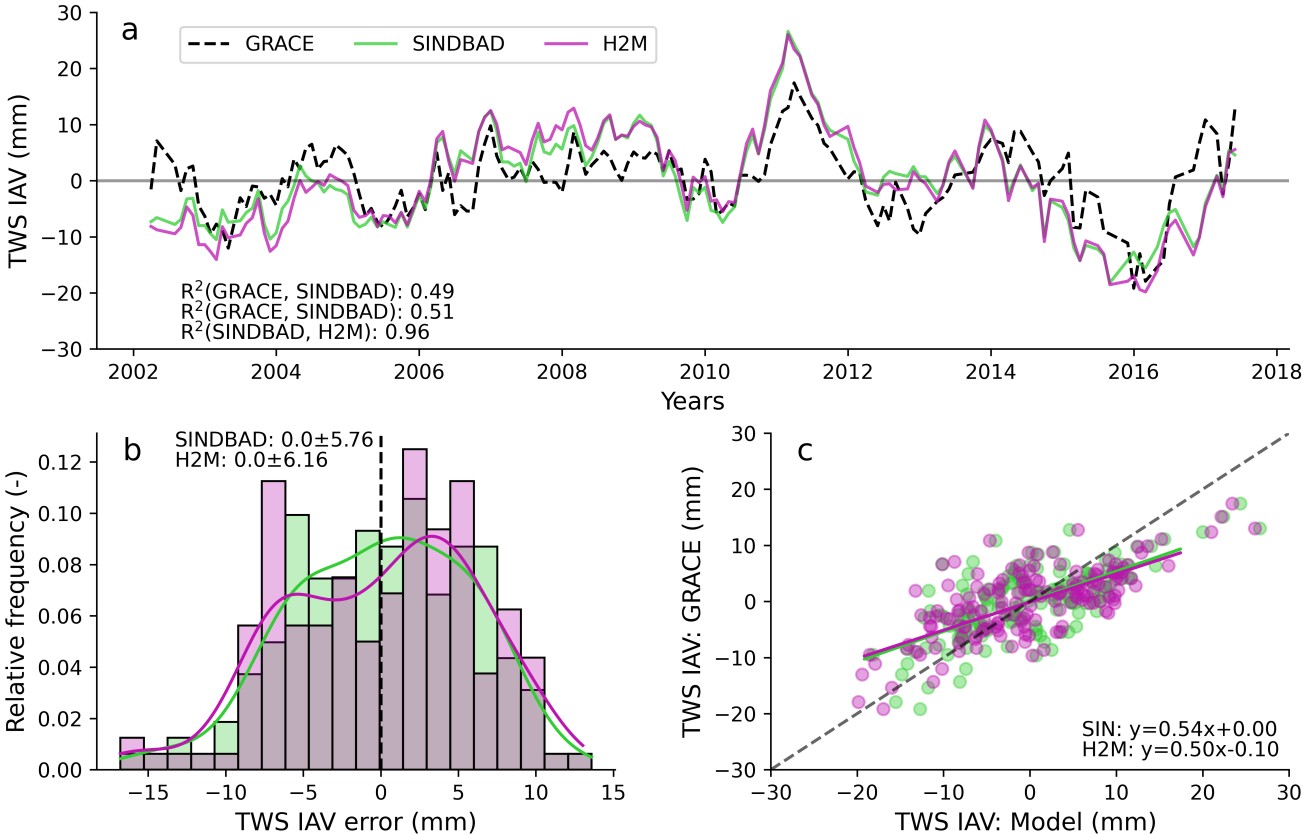

**Figure 2.** Comparison of monthly global terrestrial water storage (TWS) interannual variability (IAV) from GRACE observations and two data-driven hydrological models (SINDBAD and H2M). (a) Time series comparison of monthly global TWS IAV. $R^2$ statistics in the bottom-left is calculated as the square of the Pearson correlation coefficient; NSE is the Nash-Sutcliffe Efficiency. (b) Histogram of errors of the global TWS IAV (Eq. 3) with smoothed kernel density curves estimated using the Gaussian kernel and the Scott's rule of thumb to determine the bandwidth of the kernel. The sum of all bar heights (different models in different colors) equals unity. Shown text in the upper-left is the mean±standard deviation of the distribution of each model. (c) Scatter plot of monthly TWS IAV by GRACE and models. Equations in the bottom-right are from a robust linear regression using Huber's T estimation for downweighting outliers.

negative anomalies). Interestingly, the two models show similar temporal variations (Fig. 2a) with $R^2$ of 0.96, despite differences in modeling perspectives (i.e., process-based with calibrations against observations vs. machine learning-based estimates with only mass-balance principles). Regarding distribution of errors, the differences between estimates and observations are mostly <10 mm and normally distributed with a mean of zero, suggesting that models generally perform well (Fig. 2b).

Despite the overall good performance, the models overestimate the observed magnitude of GRACE in both tails. For instance, the models overestimated the magnitude of globally wet conditions in 2011, and dry conditions in 2015-2016 (Fig. 2a). This pattern of overestimation of magnitude is also prevalent in non-extreme years such as 2002, and 2006 to 2009, which result





in the lower $R^2$. The scatter plot between estimates and observations shows a cluster in a TWS IAV from -10 to 10 mm
(Fig. 2c). The cluster is positioned around the one-to-one line, but rather less-constrained, resulting in regression slopes 0.54
for SINDBAD and 0.50 for H2M. These slopes mean that the models overestimate (i.e., more positive) wet conditions while
they underestimate (i.e., more negative) dry conditions. The overestimation is probably due to the fact that the models lack
interactions between water storages that reduces TWS IAV. For example, surplus runoff can be redistributed within and across
pixels via lateral flow to replenish rather dry regions, which is not simulated by the models.

### 3.2    Spatial contributions to the global TWS IAV

While the models perform generally well in terms of reproducing the global TWS IAV, the differences in amplitude are signif-
icant. We conduct a covariance matrix analysis to quantify the spatial contributions of each pixel to the variance of the global
TWS IAV.

The modeled TWS IAV generally agree with the pattern from GRACE (maps along diagonal of Fig. 3) with correlation
coefficients of 0.68 and 0.73 for SINDBAD and H2M, respectively. The spatial patterns are also consistent among two models
($r$=0.89). GRACE and the models agree that the strongest positive contributions appear in the humid regions of northern South
America and Mekong River Basin, and the semi-arid regions of southern Africa, Indian subcontinent and northern Australia
(Fig. 3). On the contrary, strong negative contributions are from southeastern South America, Central Africa, and Southeastern
China.

Despite the broad similarity, there are some differences between GRACE and models. For instance, the negative contribution
of sub–Saharan regions in GRACE is not prominent in the models, and positive contributions in humid regions of Southeast
Asia and northern Australia are stronger in the models (maps above the diagonal in Fig. 3). The models tend to underestimate
the lowermost contributions and overestimate the largest contribution, which results in a slope <0.68 and intercept >0 for
the fitted lines (Fig. 3). There are also subtle differences between SINDBAD and H2M with H2M showing larger contribution
from Okavango region, and smaller contribution from northern Australia, which results in a slightly better comparison between
GRACE and H2M. In summary, GRACE and both models show globally spread out pattern of contributions to the global TWS
IAV. The difference of patterns between GRACE and models are not coherent with climate alone, as the differences are clear
in both humid and semi-arid regions.

### 3.3    Spatial contributions to the global TWS IAV error

We showed that the patterns of contributions of each pixel to the global TWS IAV are largely consistent among the models
and GRACE, but the question remains if these regions also contribute to the error in global TWS IAV. We, therefore, apply the
same covariance matrix analysis to the globally integrated TWS IAV errors (Eq. 3) to identify the regions which contribute to
the variance of the error, and identify the global error hotspots.

Despite differences in the model processes and how data were used, the two models generally agree on the distribution of the
error contributions (Fig. 4). The strongest positive contributions to the variance of error appear in the Laurentian Great Lakes
Basin, Amazon, the Paraná River lowlands, sub–Saharan regions, Indian subcontinent (Narmada, Topi, and Godavari river





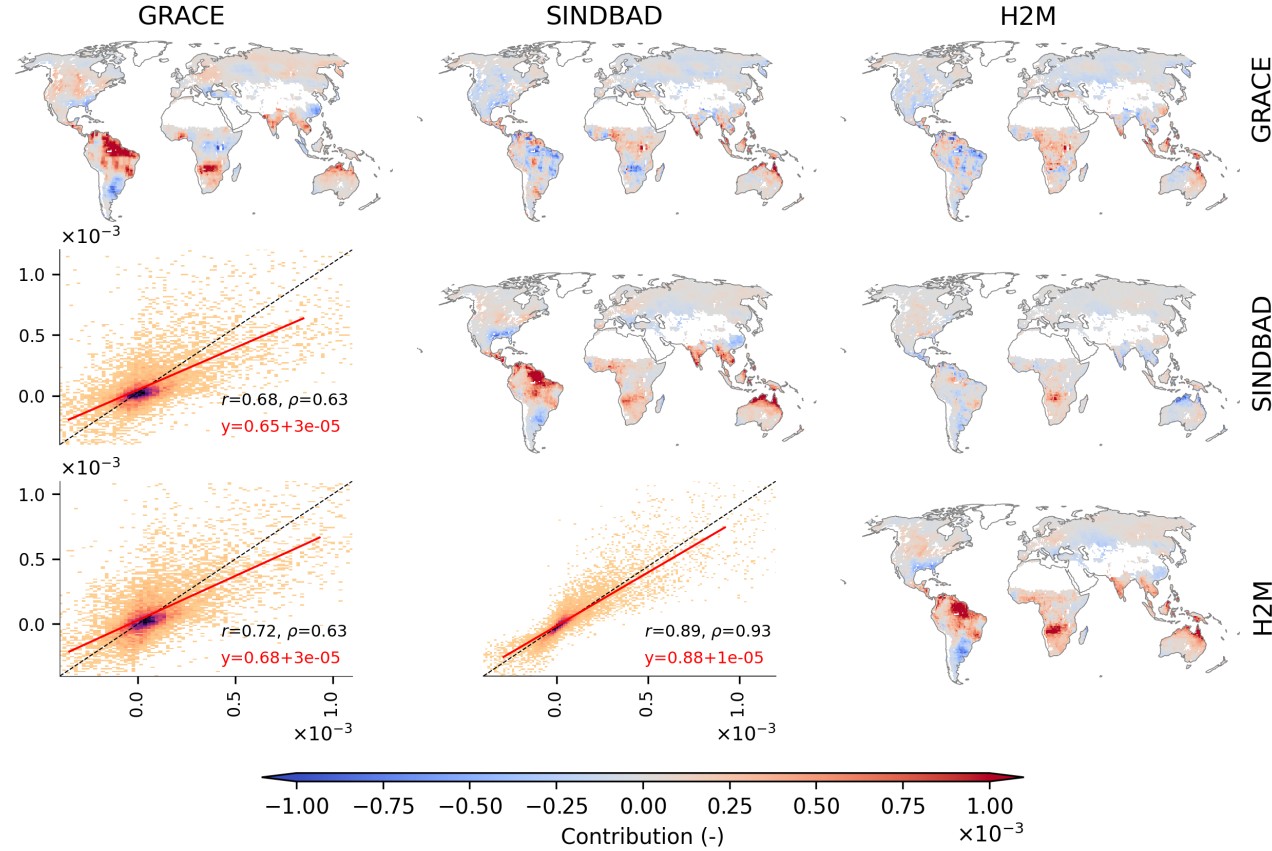

**Figure 3.** Global distribution of pixel-wise contributions to the variance of the global terrestrial water storage (TWS) IAV. Along the diagonal, maps of the pixel-wise contribution in GRACE, SINDBAD, and H2M are shown. Above the diagonal, maps of the difference (i.e., column - row) are shown. Below the diagonal, scatter plots comparing the corresponding column (x-axis) versus row (y-axis) are shown. In the scatter plots, colors indicate the density of point, $r$ is the Pearson correlation coefficient and $\rho$ is the Spearman correlation coefficient. Red lines are linear regression fit and red texts are corresponding equations.

basins), Southeast Asia, and northern Australia. For H2M, areas of positive contributions mostly coincide with areas showing strong errors in TWS IAV simulation reported by Kraft et al. (2022). Negative contributions appear in the semi-arid regions

of northeastern South America, western North America, and eastern Europe. The contribution of a pixel is barely related with the relative uncertainty of GRACE observations (Fig. B3), implying other causes of the modeling error. The distribution of contributions is centered around zero, but with a longer positive tail (histogram of Fig. 4). This suggests that a few pixels have a relatively larger positive contribution to the variance of global TWS IAV errors. We, therefore, identify these influential pixels, hereafter, the error hotspots, which have the largest influence on the change in 1) the contributions to the variance of

global TWS IAV errors and 2) $R^2$ between models and GRACE.





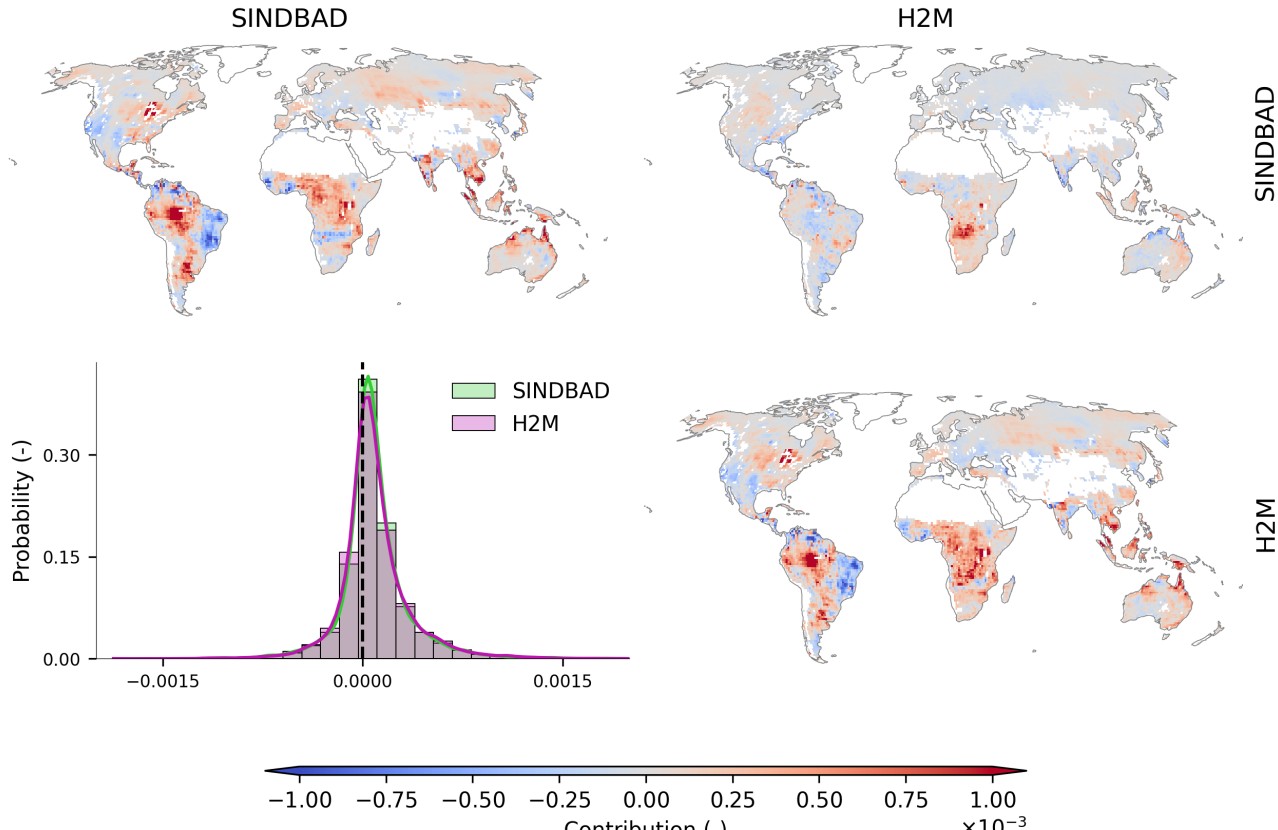

**Figure 4.** Global distribution of pixel-wise contributions to the variance of the modeling error of global terrestrial water storage interannual variability. Along the diagonal, maps of the pixel-wise contribution to the global TWS IAV modeling errors in SINDBAD and H2M are shown. Above the diagonal, a map of the difference (i.e., column - row) is shown. Below the diagonal, a histogram comparing the corresponding column (x-axis) versus row (y-axis) is shown. The probability density curves were estimated using the Gaussian kernel and the Scott's rule of thumb to determine the bandwidth of the kernel.

When the hotspot pixels are trimmed, we indeed find that the model performance improves and the error contribution to the variance of the global TWS IAV error reduces (Fig. 5). For example, the pixels with the largest 10% error contribution explain 70% of the variance of the global TWS IAV error. At the same 10% trimming, $R^2$ for global TWS IAV improves from 0.49 to 0.70 for SINDBAD and from 0.51 to 0.77 for H2M. Note that the model improvement seizes or reverses when larger percentiles are trimmed because of the loss of variance signal in the data. The results, though, confirm that a small portion of pixels (10%) is highly influential to model simulation errors at the global scale, and thereby those pixels can be regarded as hotspots of the global TWS IAV errors. In fact, models perform much worse in error hotspots ($R^2$ ranges from 0.00 to 0.52 across model-region combinations, Fig. 6) than they perform at the global scale (Fig. 2); models show much better performances in non-hotspots



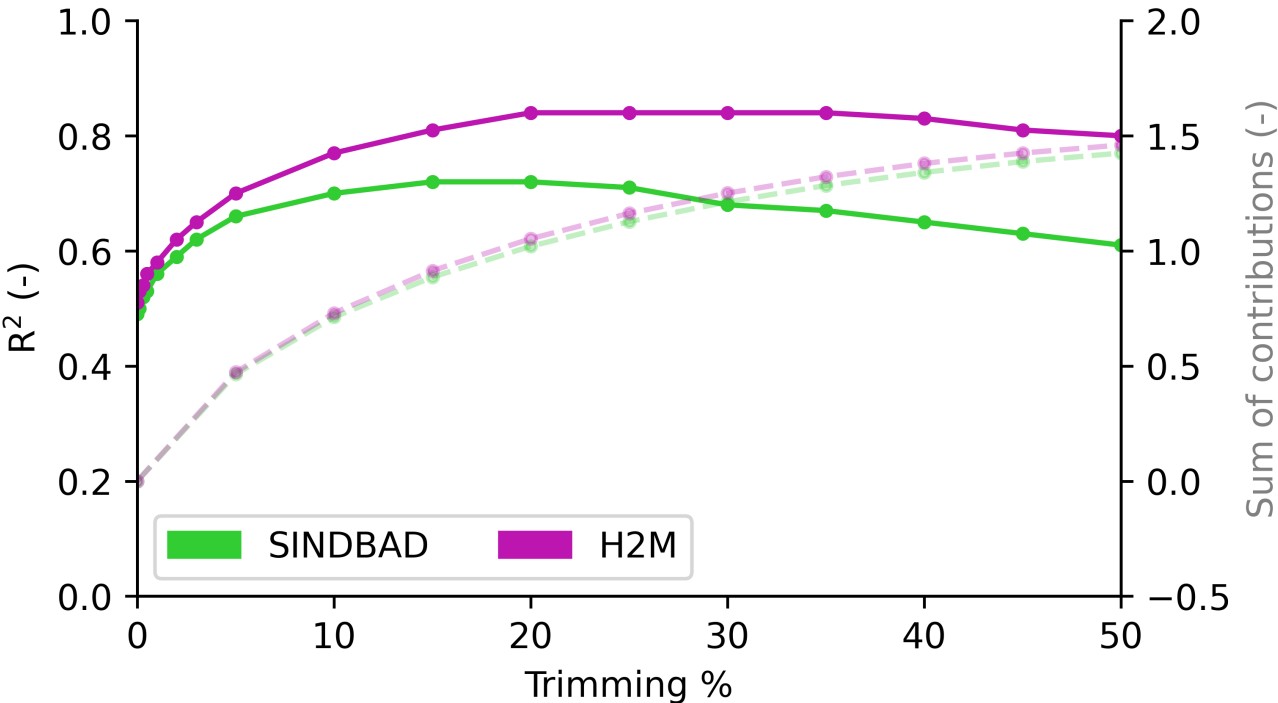

**Figure 5.** Changes in model performance and errors when pixels with largest contribution to the variance of global TWS IAV errors are trimmed. Solid lines show changes in $R^2$; dashed lines show the sum of contributions of corresponding trimmed-out pixels.

($R^2$ ranges from 0.54 to 0.95 across model-region combinations, Fig. B4) than they do in error hotspots. On a broad scale,

the error hotspots, mostly in tropical regions (Fig. B2), are common between the two models, suggesting common potentially missing mechanisms. At the regional scale, the error hotspots of each model (and their intersection) may differ slightly.

Note that the hotspot regions for error are not necessarily fully consistent with regions with the largest contribution to the global TWS IAV, as the contributions have low spatial correlation (Fig. B5). While the global TWS IAV signal is mainly dominated by moisture-limited semi-arid regions (Fig. 3), most of the error hotspots are identified in tropical regions, which

means that the regions with large contributions to the variance of global TWS IAV are not necessarily the same as those that contribute to the error in the variance of global TWS IAV (Fig. B5). In addition, the error hotspots are located over large river networks, wetland, or around lakes, which suggests possible role of surface water dynamics in TWS variability.

### 3.4   Temporal variation of TWS and climate in error hotspots

To characterize the hotspot regions and evaluate if TWS errors are systematic across time, we present the temporal variation of

TWS IAV and precipitation IAV at the regional scale for different error hotspots within selected SREX regions (see Fig. B1 for





the SREX regions). Note that the temporal variations of other climatic variables for the same regions are shown in supporting information (Figs. B6 and B7).

Across error hotspots of all selected SREX regions, modeled TWS IAV shows stronger correlations with precipitation than GRACE suggesting that the processes, such as lagged storage variations, that are not directly related to annual precipitation,

play a key role in TWS IAV. The pattern is consistent even when an independent precipitation product is used (Fig. B8). In addition, while models overestimate the range (or amplitude) of global TWS IAV (Fig. 2), the regional scale errors are bidirectional. For example, In the Laurentian Great Lakes (Fig. 6a), both positive and negative peaks (e.g., 2005–2006 and 2010–2014) are underestimated by the models. In the Amazon (Fig. 6b) as well, the peaks of TWS in drought (2010) and floods (e.g., 2009) (Chen et al., 2010) are not well captured.

**3.5 Association with hydrological variables**

While we have identified the hotspots of global TWS IAV errors, potential reasons and sources of the errors in these regions are yet not clear. To further understand how the models, which actually were forced by climate variables, could not simulate the TWS IAV well in these regions, we contrast the distribution of hydrological variables in error hotspots and non-hotspots to infer possible hydrological mechanisms behind the errors. As the model simulations were forced with climate and parameters

were constrained with the evapotranspiration and runoff data (see Appendix A), we focus on the variables that are reflective of the missing processes in the selected models. In particular, we focus on the missing processes of surface water dynamics and local and regional scale lateral moisture convergence, which may result in biases in runoff and ET processes, and consequently affects the TWS variability. Note that the association with the climate variables, which form the necessary predicate for TWS variability in the model simulations are presented in Fig. B9.

The error hotspots show a systematic larger probability density of larger maximum river storage (Fig. 7), which indicates a possible relevance and significance of river storage for the TWS interannual variability in the error hotspots. The two selected models do not explicitly account for the river or floodplain storage that accumulates large amounts of lateral flow of flood water. This means that the model would not account for the runoff loss to be further included in the delayed TWS variation, as one would expect in large river basins. Furthermore, error hotspots show larger fractions of groundwater-driven wetlands (Fig.

7) while they show smaller fractions of regularly flooded wetlands (Fig. B10) suggesting a role of surface water-groundwater dynamics as well. The dominance of groundwater-driven wetlands in error hotspots indicates that the water table depth in error hotspots is rather shallow, which forms necessary conditions for a strong interaction among soil water, groundwater, and surface water. On the other hand, spatial and temporal fraction of surface water existence do not characterize error hotspots from non-hotspots (Fig. B11). This rather unclear difference in surface water existence compared to river water storage may

be related to 1) very local occurrences (compared to large global area) of surface water and/or 2) lack of quantitative measure of water storage in the patterns of surface water existence. For example, the surface water may exist in most years in both the hotspots and non-hotspots, whereas the magnitude of existence (e.g., the number of months) can still be different but this cannot be distinguished from the given data.



**Figure 6.** Time series of 12-month moving averages of GRACE and modeled terrestrial water storage (TWS) interannual variability (IAV) in error hotspots that are commonly identified by SINDBAD and H2M (Fig. B2) within each selected region: the Laurentian Great Lakes (a, region 5), Amazon (b, region 7), Eastern and Western Africa (c, regions 15 and 16), and South Asia (d, region 23). SREX regions are shown in Fig. B1. Monthly values are shown in faded colors.

The error hotspots are also common in monsoon regions, where a clear seasonal variation of precipitation amplifies the role
of secondary moisture sources on evapotranspiration processes (Koirala et al., 2014) that consequently determines the TWS dynamics. In such regions, moisture supply from both local and remote groundwater sources have been identified as important secondary sources for vegetation transpiration during dry seasons (Miguez-Macho and Fan, 2021). Therefore, we hypothesize



**Figure 7.** Comparison of probability density distributions of the log-transformed maximum river water storage (left), and wetlands fraction (right) between the error hotspot pixels and non-hotspot pixels. Top and middle lows are distributions of each model; the bottom row is the difference in bar heights between hotspot and non-hotspot (positive means occurrences are larger in error hotspots). The probability density curves were estimated using the Gaussian kernel and the Scott's rule of thumb to determine the bandwidth of the kernel. Asterisks (*) beside model names show the significance of the difference in distributions between error hotspots and non-hotspots using the Kolmogorov-Smirnov two-sample test; all results show significant difference in distributions (***, $p$-value < 0.001). Note that the x-axis is normalized using the maximum and minimum of variables so that the range becomes zero to one, and comparisons can be made across variables. The river water storage (wRiver$_{max}$) was calculated using the Total Runoff Integrating Pathways (TRIP) river routing model (Oki and Sud, 1999) with the input of runoff from SINDBAD. The maximum wRiver of a pixel (wRiver$_{max}$) during the entire period (April 2002–June 2017) was used with log transformation to use the skewed distribution of (wRiver$_{max}$) for the comparison. The fraction of groundwater-driven (GW-driven) wetlands was provided by Tootchi et al. (2019).





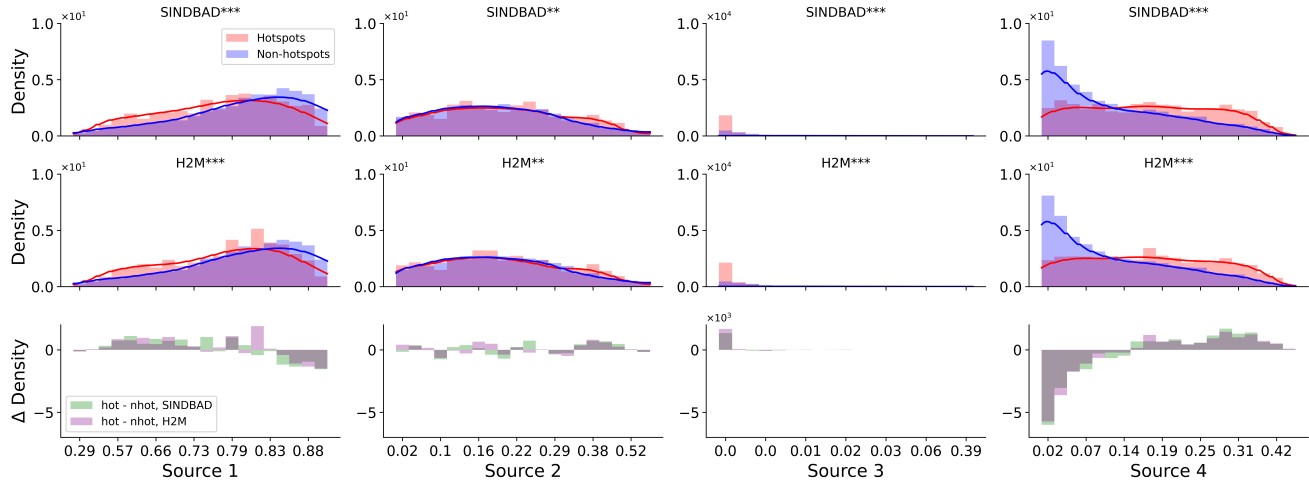

**Figure 8.** Same as Fig. 7, but for the contribution of four water sources to the water usage by vegetation. Data by Miguez-Macho and Fan (2021) was used for the four sources. Source 1 is soil water from recent (< 1 month) precipitation; source 2 is soil water from past precipitation; source 3 is locally-recharged groundwater via capillary flow; source 4 is remotely recharged groundwater from uplands to lowlands.

that the error hotspots are more prevalent in regions where contribution of secondary moisture sources is significant. To test this hypothesis, we evaluate the associations of global TWS IAV error hotspots with groundwater uptake by vegetation using

an existing dataset of transpiration sources (Miguez-Macho and Fan, 2021).

We find that the probability of vegetation accessing the secondary moisture is much larger in the error hotspots of TWS IAV. In particular, this association is stronger when the secondary groundwater moisture is coming from the upstream regions (source 4, Fig. 8). The hotspot occurrences are larger in regions where the primary source of moisture only supports a relatively smaller fraction of transpiration (source 1, Fig. 8). A larger fraction of transpiration here, therefore, may be supported by secondary

and non-trivial sources. We find that the difference between occurrences of error hotspots and non-hotspots is small in regions where transpiration is dependent on locally-recharged groundwater (i.e., source 3). On the other hand, the difference is clearer in the regions where transpiration is supplied by remotely recharged groundwater (i.e., source 4).

## 4 Discussion

Getting back to the three research questions, we first discuss the regions most contributing to the global TWS IAV and its

modeling error (Sect. 4.1), then finally we discuss potential sources of the error (Sect. 4.2).





## 4.1 Spatial contributions to the global TWS IAV and its error

The semi-arid regions are main contributors to the variance of the global TWS IAV (Fig. 3), which agrees with the findings of previous studies. For example, Humphrey et al. (2018) showed a similarly large role of semi-arid regions to the global TWS IAV, but found that only GRACE shows a large contribution of tropical regions. In that regard, the use of GRACE data in

SINDBAD and H2M, perhaps, helps produce a regional and global pattern of TWS that is more consistent with GRACE than previously reported. As the semi-arid regions also have a dominant role in IAV of the global terrestrial carbon sink (Poulter et al., 2014; Ahlstrom et al., 2015; Jung et al., 2017; Humphrey et al., 2018), the large contributions of semi-arid regions to the global TWS IAV implies a strong global-scale linkage between the water and carbon cycles (Law et al., 2002), as shown in Humphrey et al. (2018). But, the role of the large TWS IAV contribution of humid tropical regions, which is prevalent in

GRACE and the two data-driven models selected here, and its effect on the linkages of global water and carbon cycle still remains to be understood.

On the other hand, tropical regions come out as the dominant contributor to the variance of the global TWS IAV modeling errors (Fig. 4). Tropical regions were reported as one significant contributor to the global TWS IAV, but with a large disparity between the models and GRACE (Humphrey et al., 2018). While SINDBAD and H2M mostly agree on the distribution of

contributions, the two models show differences particularly in the semi-arid regions of southern Africa, where SINDBAD shows a negative contribution, while H2M shows mostly positive contribution, resulting in a large positive difference between the models. The difference could possibly be attributable to whether a model considers the capillary rise (SINDBAD does and H2M does not) which changes the TWS dynamics, as vegetation can access and lose water as transpiration in the dry season (Guan et al., 2014; Madani et al., 2020; Miguez-Macho and Fan, 2021). In SINDBAD, vegetation indirectly accesses secondary

water storage with capillary rise, which contributes to a larger evapotranspiration over some regions, including the regions in Africa (Fig. 9 in Trautmann et al., 2022).

## 4.2 Potential sources of errors

In the comparison of TWS IAV and precipitation IAV time series within selected SREX regions in error hotspots, we find that modeled TWS IAV is more tightly correlated with precipitation IAV than GRACE TWS IAV is (Fig. 6). The consistent quicker

response of TWS IAV in the models points to a lack of 1) sub-pixel and across pixel moisture convergences or 2) insufficient representation of delayed storage which is either simply represented in SINDBAD or ignored in H2M. The inferences were well supported by the characteristics of the error hotspots such as a larger river water storage, frequent occurrences of groundwater-driven wetlands, and a stronger contribution of groundwater recharged via the uplands-to-lowlands to transpiration (Figs. 7 and 8). The error hotspots are located in regions with a large potential of lateral flow of the surface water, e.g, regions with a well

developed river network (e.g., Amazon, as shown in Jung et al., 2010), or with a large accumulation of snowmelt in spring (e.g., Laurentian Great Lakes basin as shown in Xiao et al., 2016; Huziy and Sushama, 2017). In these regions, the lateral flow delays responses of TWS to precipitation (Soni and Syed, 2015), effectively nullifying their correlation as seen in GRACE observation. The lack of or the underrepresentation of surface or delayed water dynamics is pointed out as a critical improvement for the





future (Kraft et al., 2022). SINDBAD represents a conceptually lumped delayed water storage (i.e., groundwater and surface
water) using a state variable, wSlow. The wSlow has an unlimited storage capacity that depends on a globally constant model
parameter, but it does not have spatio-temporal variability. Therefore, the lumped wSlow representation may not be sufficient
to reproduce globally varying but locally relevant large contribution of surface water storage to TWS IAV (e.g., Kim et al.,
2009; Frappart et al., 2012; Getirana et al., 2017; Pokhrel et al., 2018; Schrapffer et al., 2020).

It has been reported that the water storages with delaying processes are relevant to the regional and global hydrology. River
storage explains up to 73% of TWS variability in the Amazon (Kim et al., 2009). River routing affects the water flow patterns
(Jung et al., 2010) and variability of simulated runoff (Jin et al., 2021). Improvements in TWS simulations have been reported
in Amazon, Africa, and India when river routing is considered in the model (Getirana et al., 2017). Humphrey et al. (2018)
pointed out that models are limited in representing groundwater-surface water dynamics, e.g., in wetlands, which may have a
slower or no response to climatic forcings. The lack of slow water processes such as lateral flow will cause errors in simulating
the hydrological cycle, and will further result in errors in simulating interactions among cycles, such as the water-carbon cycle
interactions (Humphrey et al., 2018; Madani et al., 2020).

Other than the difference in the strength of correlation, models also do not capture some peaks of TWS IAV in the error
hotspots. Missing water storage in models, for example, floodplain storages may be related to the smaller accumulation and
lower positive anomaly than seen in GRACE, as also reported previously (Scanlon et al., 2018) and as the error hotspots have
larger river water storages compared to non-hotspots (Fig. 7). The underestimation of the negative peaks may also be related to
the use of a small soil water storage capacity in the models, which would not allow for a depletion of deeper moisture storage
in infrequent but significant drought years. Soil water storage capacity in SINDBAD is mostly less than 2 m (Fig. B12) and
Swenson and Lawrence (2015) reported that soil water storage capacity of 8-10 m is required to replicate TWS variability over
tropical regions observed by GRACE. However, the storage capacity is probably not the only reason, as H2M, which does not
predefine soil water storage capacity, shows similar underestimations.

The error hotspots were also more prevalent in regions where the transpiration is significantly supported by the secondary
moisture sources, especially the one from non-local groundwater recharge (Miguez-Macho and Fan, 2021). Even though SIND-
BAD accounts for the groundwater usage by vegetation, both the selected models show very similar distributions of the reliance,
probably because SINDBAD also only represents the local recharge process. As the local distribution of groundwater recharge
is prevalent at the hillslope scale, the remotely recharged groundwater, calculated at 30 arc-seconds resolution, can be inter-
preted as a sub-pixel scale process for the spatial resolution of 1 ° spatial resolution used in the selected models. The missing
sub-pixel convergence of moisture and its effect on water availability and on supporting ET would result in a bias in ET simu-
lations in the selected models, which would consequently translate to erroneous TWS. Indeed, we find that the selected models
underestimate the ET compared to FLUXCOM observations (Fig. B13). On the other hand, the two models do not show a clear
pattern with biases in runoff (i.e., sub-pixel lateral flow) simulation. This suggests that in some regions local redistribution of
moisture and its effect on TWS variability is modulated by evapotranspiration rather than runoff processes.

Lastly, the additional sources of errors include anthropogenic influence and uncertainties in the GRACE data. In Africa, a
decrease in TWS IAV in 2003–2006 (Fig. 6c) was due to the expansion of Nalubaale Dam as well as La Niña (Stager et al.,





2007; Awange et al., 2013, 2019). In the Indian subcontinent (Fig. 6d), the TWS changes have been reported due to human
impacts such as reducing groundwater abstraction and surging reservoirs as well as increased precipitation (Meghwal et al.,
2019; Munagapati et al., 2021). Note that most of these regions were excluded in the analysis of model simulations (Kraft et al.,
2022; Trautmann et al., 2022), and trends were removed from all data and simulations to focus on the interannual variability of
TWS under natural conditions. Regarding the GRACE errors, the GRACE mascon solution has the leakage error created during
spatial interpolation from the original spatial resolution ($3°$) to higher resolution ($0.5°$) (Wiese et al., 2016). Additionally, the
Laurentian Great Lakes, which are visible as error hotspots, are also known to be in the effect of the post-glacial rebound
(Peltier et al., 2015). The glacial isostatic adjustment is the major source of error in GRACE signal processing (Rodell et al.,
2018).

## 5 Conclusions

We provide a comprehensive application of a covariance matrix method that is effective to attribute global variability to each
pixel. The method provides a platform to compare the magnitude and direction of contribution of a single pixel or regions com-
prising arbitrary groups of pixels, owing to the normality and additive properties of the resulting statistic. Using this method,
we quantified the contribution of each pixel to the global TWS IAV of GRACE observations and two selected predominantly
data-driven models, SINDBAD and H2M, as well as its modeling errors.

We found that the global TWS IAV is mainly driven by humid tropical and semi-arid regions. On the other hand, we identified
the hotspots of modeling errors of the global TWS IAV mainly in tropical regions that span across climatic regions. These
different driving regions for the global TWS IAV and its errors show that the regions that dominate global TWS IAV are not
necessarily the same as those that dominate the error in global TWS IAV. This allowed for a further analysis in identifying the
error hotspots and potential missing mechanisms in the models. Interestingly, we found that the largest 10% error contributors
explain 70% of the global error, showing a disproportionately large significance of a small region in influencing the global
mismatch with the observations.

Using the error hotspots as the base for comparison, the TWS-precipitation IAVs have a stronger correlation in the models
than in observation. This suggests a different temporal dynamics of TWS and precipitation in the observations, that are not
captured by the models due to missing representations of key hydrological processes in some regions, which are congregated
locally across different climate but still have a large influence on global errors. In fact, by associating with hydrological
variables, we found that the error hotspots are regions with larger river storage, lateral flow, and more contribution of lateral
flow to transpiration via recharging groundwater which points to critical but missing processes of runoff accumulation, storage
dynamics, and roles of secondary moisture in evaporative processes.

Our findings provide an improved understanding of the global TWS IAV and its modeling error. The models can be improved
for global TWS IAV by focusing on the subgrid heterogeneity of moisture convergence and lateral flow processes that dominate
the error hotspot regions. We also highlight the risk of inferring general process attributions and causations using global





associations between variables, as only a small fraction of pixels may have significant implications of interannual variabilities at the global scale.

## Appendix A: Forcings and constraints

The forcing variables included precipitation, air temperature, and net radiation, and these were retrieved from GPCP 1dd v1.3

(Huffman et al., 2001), CRUJRA v2.2 (Harris, 2021), and CERES SYN1degEd4A (Wielicki et al., 1996), respectively. Four observational data streams were used to constrain both models: 1) TWS from the Gravity Recovery and Climate Experiment (GRACE) Mascon Equivalent Water Height RL06 version 1 with Coastal Resolution Improvement (CRI) version 1 (Wiese et al., 2016), 2) snow water equivalent (SWE) from the GlobSnow v3 (Luojus et al., 2021), 3) evapotranspiration from the FLUXCOM v1 RS ensemble (Jung et al., 2019), and 4) runoff from the G-RUN Ensemble v1 (Ghiggi et al., 2021). Note that

vegetation characteristics such as vegetation index and maximum rooting depth were only used for SINDBAD, while time-static land surface characteristics such as soil properties and wetland fractions were only used for H2M. See Trautmann et al. (2022) and Kraft et al. (2022) for details of SINDBAD and H2M, respectively.

All forcing and constraints were aggregated into the 1° spatial resolution using the land area of each pixel; SWE was aggregated to the monthly scale. For GRACE, the original data has an irregular timestep and occasionally has two observations within a month with missing value for the previous or next month. Two months (January 2012 and April 2015) have two

observations for the study period (April 2002–June 2017, matching that of GRACE data). The earlier observation in January 2012 was regarded as an observation in December 2011 and the later observation in April 2015 was regarded as an observation in May 2015. The scaling factor was applied before the spatial aggregation. Gaps in the time series of each pixel were not filled, and all extrema were assumed to be a reasonable signal and included in the analysis. The GlobSnow SWE was spatially gap-filled to zero values in non-snow regions to get the global coverage following Kraft et al. (2022). For the FLUXCOM

ET, only a part of the study period was available (2002–2015). Lastly, the two models covered different land pixels due to independent data filtering. Therefore, only the common land pixels between two model simulations were used in this analysis, and the same land mask was applied to all forcing and constraints.

## Appendix B: Additional figures



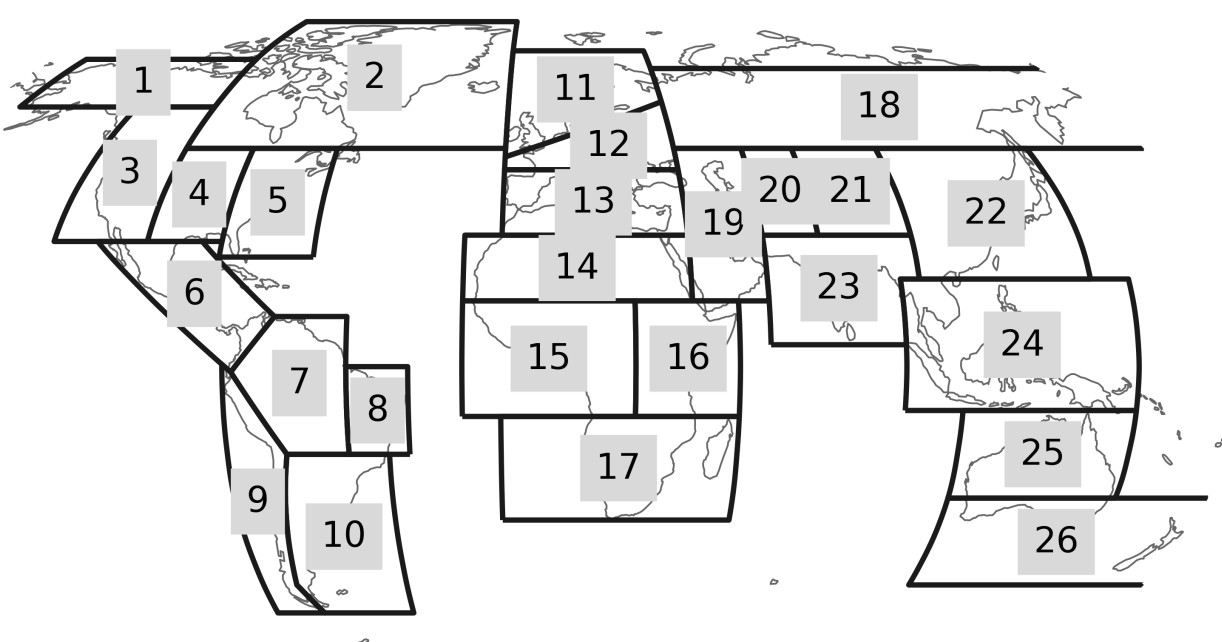

**Figure B1.** SREX regions (Seneviratne et al., 2012), which were used in figures (Figs. 6, B4, B6, B7, and B8) to spatially average global terrestrial water storage interannual variability time series into four selected regions: the Laurentian Great Lakes (SREX regions 5), Amazon (SREX regions 7), Eastern and Western Africa (SREX regions 15 and 16), and South Asia (SREX regions 23).





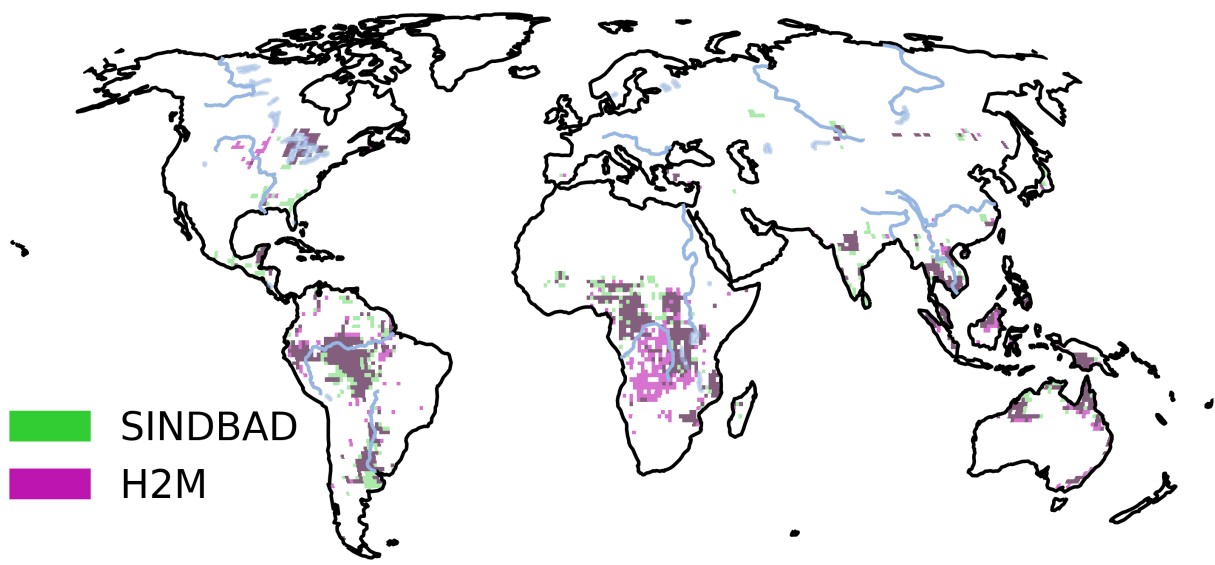

**Figure B2.** Spatial patterns of error hotspots of modeling errors in terrestrial water storage (TWS) interannual variability (IAV). Error hotspots mean pixels with the 10% largest positive contributions to the global TWS IAV modeling errors (Fig. 4). Red pixels are error hotspots identified using SINDBAD; light purple pixels are error hotspots identified using H2M; and dark purple pixels are commonly identified error hotspots by the two models.





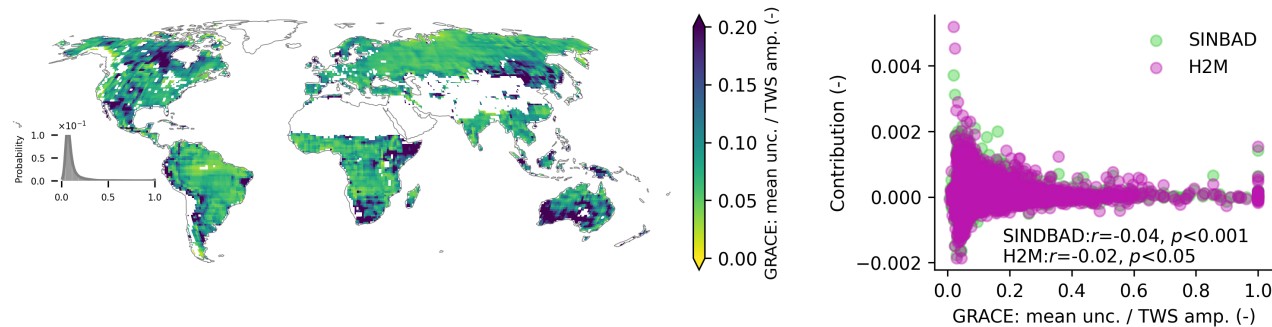

**Figure B3.** (left) Spatial distribution of the relative uncertainty of GRACE terrestrial water storage (TWS) observations. The relative uncertainty of each pixel is calculated as the mean uncertainty divided by the amplitude of TWS. (right) Scatter plots comparing the relative uncertainty of GRACE TWS (x-axis) versus pixel-wise contributions to the global TWS IAV modeling error (y-axis). Shown below as text is the Pearson correlation coefficient with $p$-value of each model.





**Figure B4.** Same as Fig. 6, but using non-hotspot pixels.



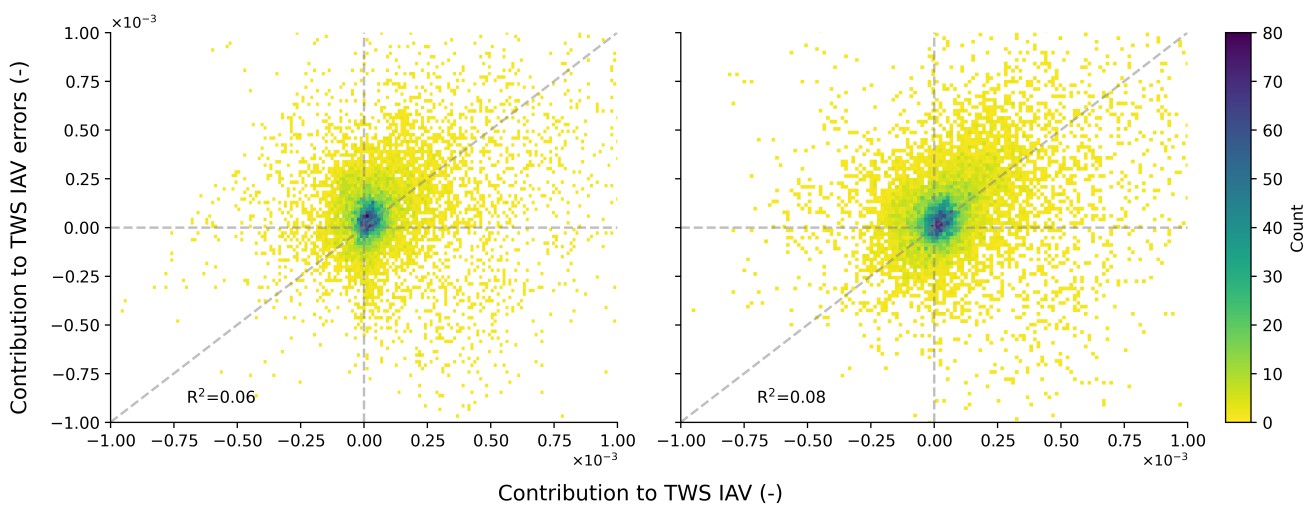

**Figure B5.** Bivariate histograms that compare pixel-wise contribution to the global terrestrial water storage (TWS) interannual variability (IAV) versus pixel-wise contributions to the global TWS IAV modeling error in SINDBAD (left) and H2M (right). The pixel-wise contribution is quantified using Eq. 4 (See Sect. 2.1.2). Colors mean the count of cases of corresponding ranges of contribution to TWS IAV and contribution to TWS IAV errors. $R^2$ at the bottom-left is calculated as the square of Pearson correlation coefficient.

**Figure B6.** Same as Fig. 6, but using air temperature (Tair) IAV instead of PPT IAV.

**Figure B7.** Same as Fig. 6, but using net radiation (Rn) IAV instead of PPT IAV.





**Figure B8.** Same as Fig. 6, but using MSWEP, instead GPCP1dd, for PPT IAV.





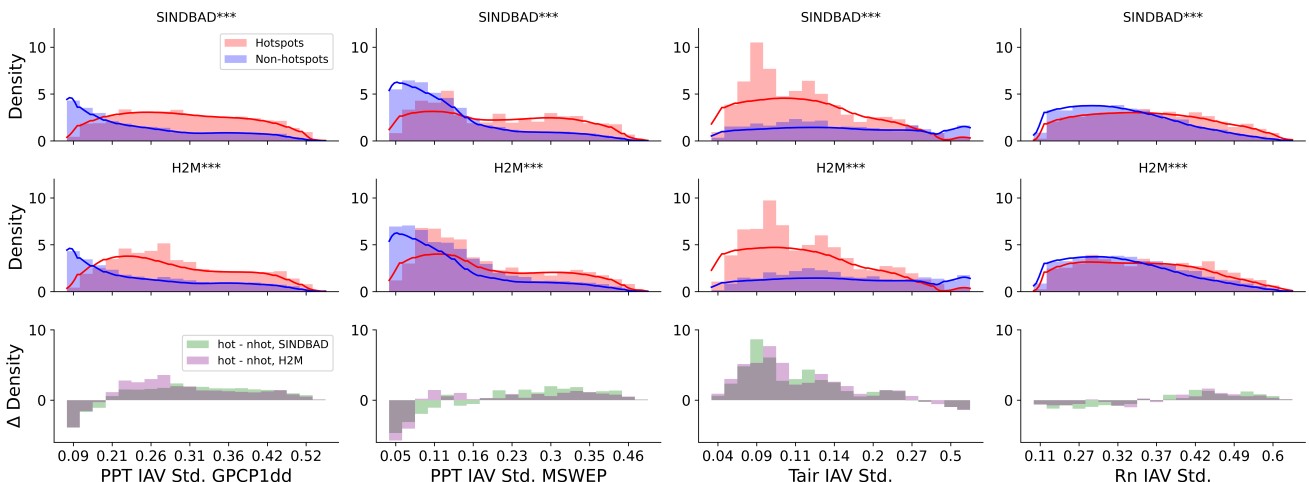

**Figure B9.** Same as Fig. 7, but for meteorological forcing variables: standard deviation (std) of precipitation (PPT) interannual variation (IAV), air temperature IAV std, and net radiation (Rn) IAV std. The first two columns from the left show results using different PPT data sets (GPCP1dd and MSWEP).





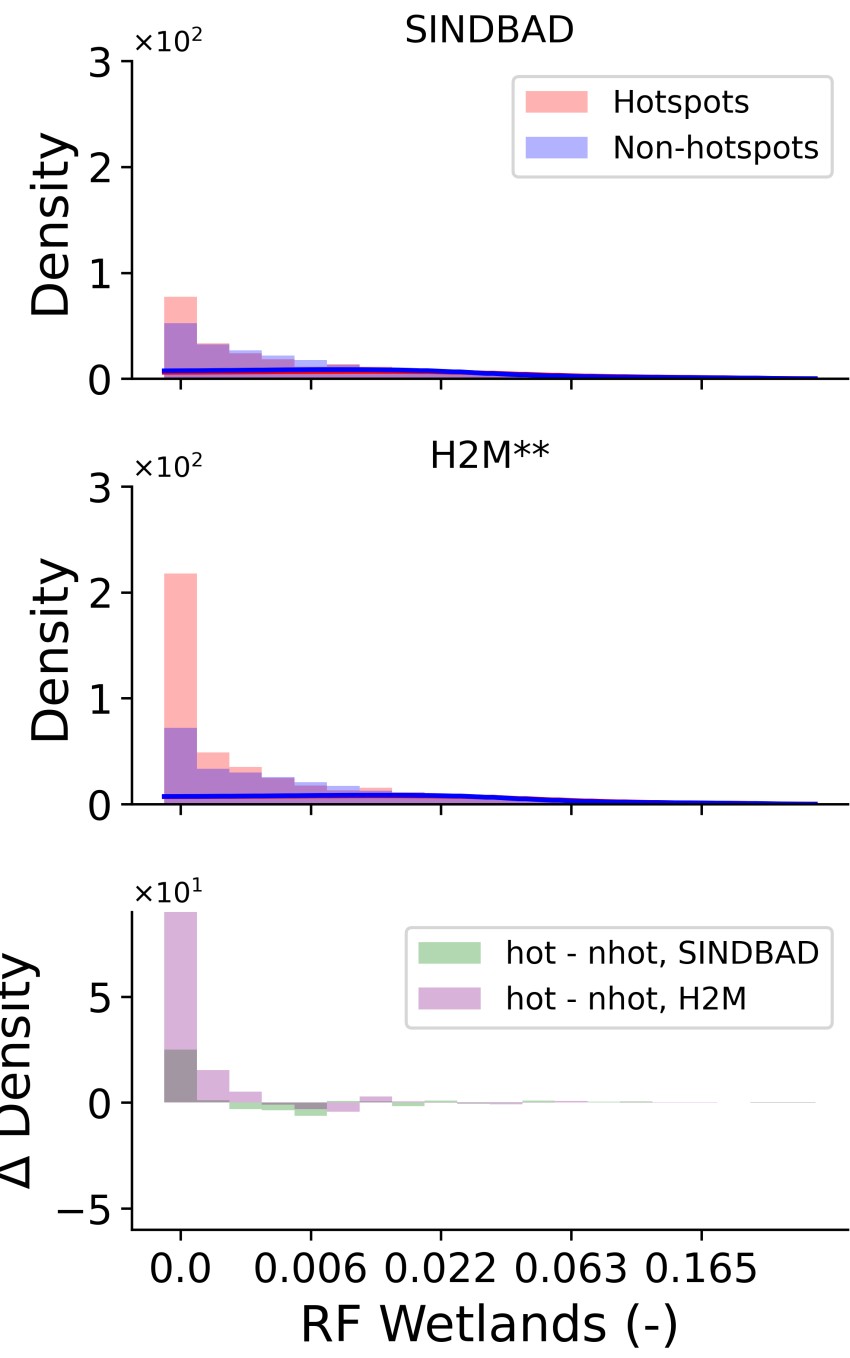

**Figure B10.** Same as Fig. 7, but for the fraction of regularly flooded (RF) wetlands provided by Tootchi et al. (2019).

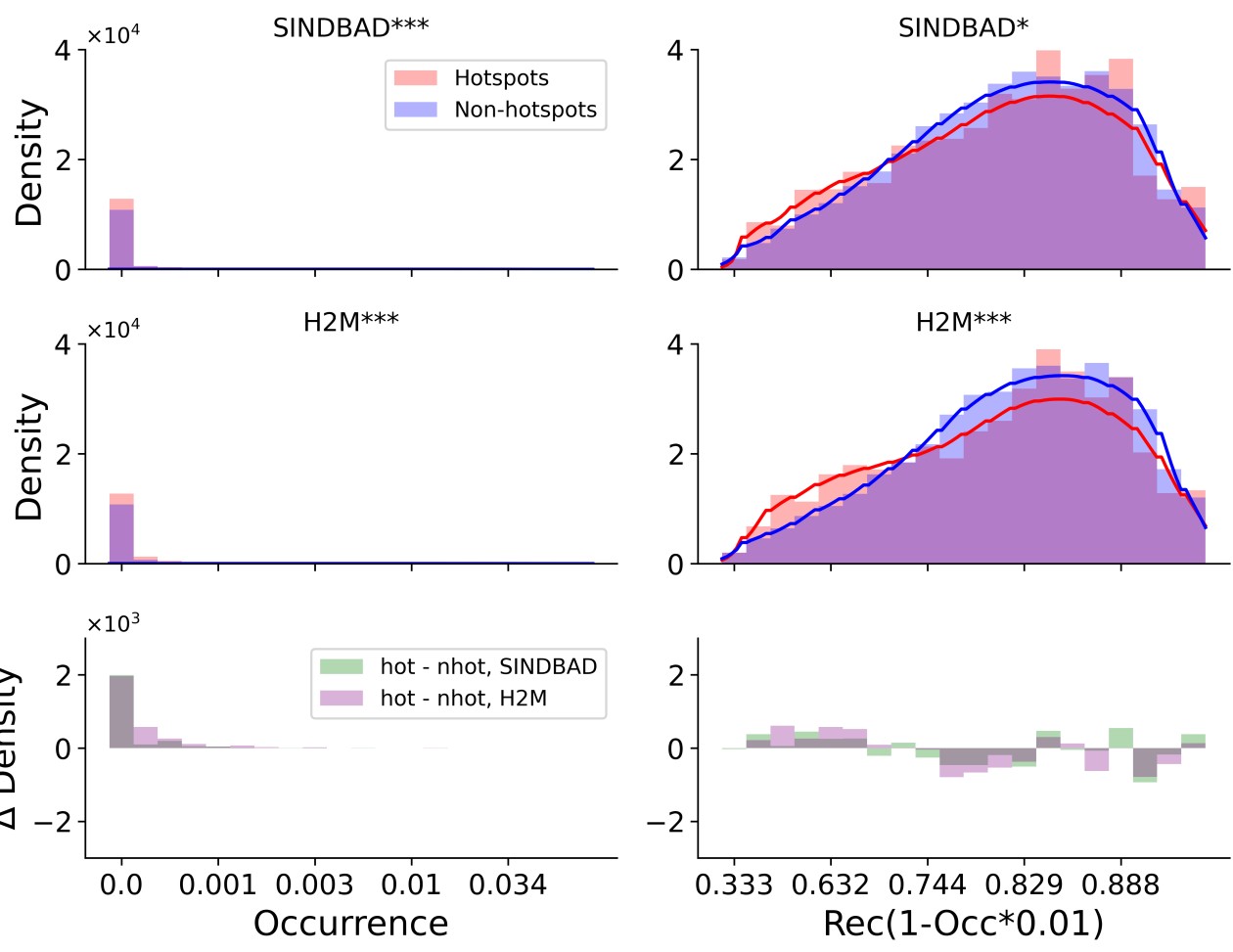

**Figure B11.** Same as Fig. 7, but using occurrence (left), the product of recurrence and occurrence (right) of the surface water bodies from global surface water explorer data set (GSWE, Pekel et al., 2016).





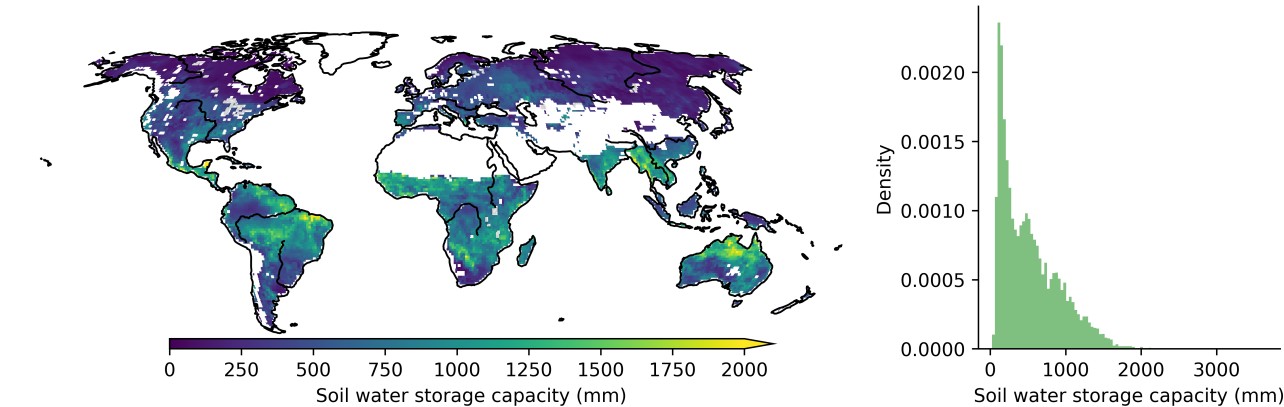

**Figure B12.** Spatial distribution (left) and histogram (right) of soil water storage capacity of SINDBAD, as the sum of capacity of two soil layers.

**Figure B13.** Same as Fig. 7, but for mean bias in evapotranspiration (ET, left column) and runoff (right column) simulations by two tested models. Note that the x-axis is not in the normalized unit using minimum and maximum but in the original unit.



*Author contributions.* MJ, SK, and HL designed the experiments. TT and BK performed the model simulations. In close collaboration with SK, HL conducted the analysis and prepared the first draft of the manuscript. All authors contributed to the research discussions, and improving the manuscript.

*Competing interests.* The authors declare that they have no conflict of interest.

*Acknowledgements.* Hoontaek Lee acknowledges support from the Max Planck Institute for Biogeochemistry (MPI-BGC) and the International Max Planck Research School for Global Biogeochemical Cycles (IMPRS-gBGC). We also thank Uli Weber at MPI-BGC for the collection and preparation of the data used in the model simulations and the analysis.



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
