# Peer review of "Diagnosing modeling errors of global terrestrial water storage interannual variability"

_Hydrology and Earth System Sciences, 2022_

## Author Comment (AC1)

**Author's response to Referee #1**

This study quantified the contribution of each pixel to the global TWS IAV of GRACE observations and two selected predominantly data-driven models, SINDBAD and H2M, as well as its modeling errors. The results show that the global TWS IAV is mainly driven by humid tropical and semi-arid region. The hotspots of modeling errors of the global TWS IAV are mainly located in tropical regions that span across climatic regions. The study provides an improved understanding of the global TWS IAV and its modeling error. Generally, the topic is important, and the study is well written and easy to follow. My comments are as follows.

**AC**: We would like to thank the reviewer for positive feedback on the study, as well as for the suggestions to improve it further. We address the comments here, and will include the changes in the revised submission.

**1. In the high latitudes of the northern hemisphere, glacier changes contribute to TWS, whether the SINDBAD model and the H2M model have a glacier module.**

**AC**: Though glacier changes significantly contribute to TWS, especially to its trend (Rodell et al., 2018; Scanlon et al., 2018), in the high latitudes of the northern hemisphere, the two models, SINDBAD and H2M, do not consider the contribution or process. To account for this limitation, grid cells with > 10% of permanent snow and ice cover were excluded from the study area (Kraft et al., 2022; Trautmann et al., 2022). We will clarify it further in the revision.

**2. It needs to be further pointed out that the model is inconsistent with GRACE in typical irrigation areas, such as the western United States, northern India, etc.**

**AC**: We agree about the potential inconsistency between GRACE and models in typical irrigation areas. That is why the analysis excludes regions with the largest anthropogenic influence on TWS trends according to Rodell et al. (2018). We will address this aspect in the manuscript as:

[...] Lastly, the two models covered different land pixels due to independent data filtering (Kraft et al., 2022; Trautmann et al., 2022). The data filtering excludes land pixels with 1) significant fraction of ice, snow, water body, bare land surface or artificial land cover, or 2) a large human influence on trend in GRACE TWS mainly by groundwater extraction. Therefore, only the common land pixels between two model simulations were used in this analysis, and the same land mask was applied to all forcing and constraints.

We will clarify the potential inconsistency also in the discussion as:

Lastly, the additional sources of errors include anthropogenic influence and uncertainties in the GRACE data. Though the analysis excludes pixels with a large anthropogenic influence

on TWS such as Northern India following Rodell et al. (2018), the human impact still remains out of the excluded pixels. In Africa, a decrease in TWS IAV in 2003–2006 (Fig. 6c) was due to the expansion of Nalubaale Dam as well as La Niña (Stager et al., 22 https://doi.org/10.5194/hess-2022-284 Preprint. Discussion started: 5 August 2022 c Author(s) 2022. CC BY 4.0 License. 2007; Awange et al., 2013, 2019). In the Indian subcontinent (Fig. 6d), the TWS changes have been reported due to human 485 impacts such as reducing groundwater abstraction and surging reservoirs as well as increased precipitation (Meghwal et al., 2019; Munagapati et al., 2021). In addition, typical irrigation areas such as the Corn Belt in the USA and northern India (Fig. 4) [...]

**3. Figure 2(a) shows that the two models are in good agreement, and they both have some differences from GRACE. Does the input of precipitation significantly affect the simulation results of the model? If other precipitation products are used as input, will the results be different?**

**AC**: Thank you for raising an important point. In the original manuscript, we evaluated the association of TWS variability with different precipitation products and found little influence on the results (Fig. B8 and B9 in the manuscript).

Nevertheless, we have tested the robustness of the results of SINDBAD and H2M by forcing the models using an independent precipitation estimate from MSWEP v.2.8 (Table 1 in the manuscript) instead of GPCP1dd, and repeated the whole analysis. We could again verify that using another precipitation forcing does not significantly change the results and findings of our study. We will highlight this finding in the manuscript and include the relevant analyses (Fig. 1~5, below) in the appendix.

Specifically, two models are still in good agreement when they are forced by MSWEP. The performance of two models has been slightly improved, shown by  $R^2$ , the distribution of errors, and the slope of regression equations (Fig. 1). However, the main findings and conclusions are not affected by that and stay the same. For example, spatial contribution to TWS IAV (Fig. 2) and its modeling error (Fig. 3) remain similar in general, so do the systematic larger wRivermax (Fig. 4) and more contribution of remotely-recharged groundwater to transpiration (Fig. 5) in error hotspots.

This is in line with Kraft et al. (2022) who showed that H2M gave almost the same performance when it is trained on different precipitation products.

Figure 1. **The same as Fig. 2 in the manuscript, but using MSWEP for the precipitation forcing.** Comparison of monthly global terrestrial water storage (TWS) interannual variability (IAV) from GRACE observations and two data-driven hydrological models (SINDBAD and H2M). (a) Time series comparison of monthly global TWS IAV. *R*2 statistics in the bottom-left is calculated as the square of the Pearson correlation coefficient. (b) Histogram of errors of the global TWS IAV (Eq. 3) with smoothed kernel density curves estimated using the Gaussian kernel and the Scott's rule of thumb to determine the bandwidth of the kernel. The sum of all bar heights (different models in different colors) equals unity. Shown text in the upper-left is the mean±standard deviation of the distribution of each model. (c) Scatter plot of monthly TWS IAV by GRACE and models. Equations in the bottom-right are from a robust linear regression using Huber's T estimation for downweighting outliers.

---

## Author Response (AR1)

**Author's response to Referee #1**

**This study quantified the contribution of each pixel to the global TWS IAV of GRACE observations and two selected predominantly data-driven models, SINDBAD and H2M, as well as its modeling errors. The results show that the global TWS IAV is mainly driven by humid tropical and semi-arid region. The hotspots of modeling errors of the global TWS IAV are mainly located in tropical regions that span across climatic regions. The study provides an improved understanding of the global TWS IAV and its modeling error. Generally, the topic is important, and the study is well written and easy to follow. My comments are as follows.**

**AC**: We would like to thank the reviewer for positive feedback on the study, as well as for the suggestions to improve it further. We address the comments here, and will include the changes in the revised submission.

**1. In the high latitudes of the northern hemisphere, glacier changes contribute to TWS, whether the SINDBAD model and the H2M model have a glacier module.**

**AC**: Though glacier changes significantly contribute to TWS, especially to its trend (Rodell et al., 2018; Scanlon et al., 2018), in the high latitudes of the northern hemisphere, the two models, SINDBAD and H2M, do not consider the contribution or process. To account for this limitation, grid cells with > 10% of permanent snow and ice cover were excluded from the study area (Kraft et al., 2022; Trautmann et al., 2022). We will clarify it further in the revision.

**2. It needs to be further pointed out that the model is inconsistent with GRACE in typical irrigation areas, such as the western United States, northern India, etc.**

**AC**: We agree about the potential inconsistency between GRACE and models in typical irrigation areas. That is why the analysis excludes regions with the largest anthropogenic influence on TWS trends according to Rodell et al. (2018). We will address this aspect in the manuscript as:

*[...] Lastly, the two models covered different land pixels due to independent data filtering* *(Kraft et al., 2022; Trautmann et al., 2022). The data filtering excludes land pixels with 1) significant fraction of ice, snow, water body, bare land surface or artificial land cover, or 2) a large human influence on trend in GRACE TWS mainly by groundwater extraction.* *Therefore, only the common land pixels between two model simulations were used in this analysis, and the same land mask was applied to all forcing and constraints.*

We will clarify the potential inconsistency also in the discussion as:

*Lastly, the additional sources of errors include anthropogenic influence and uncertainties in the GRACE data.* *Though the analysis excludes pixels with a large anthropogenic influence*

*on TWS such as Northern India following Rodell et al. (2018), the human impact still*
*remains out of the excluded pixels.* In Africa, a decrease in TWS IAV in 2003–2006 (Fig. 6c)
was due to the expansion of Nalubaale Dam as well as La Niña (Stager et al., 2007;
Awange et al., 2013, 2019). In the Indian subcontinent (Fig. 6d), the TWS changes have
been reported due to human impacts such as reducing groundwater abstraction and surging
reservoirs as well as increased precipitation (Meghwal et al., 2019; Munagapati et al.,
2021). *In addition, typical irrigation areas such as the Corn Belt in the USA and northern*
*India (Fig. 4)*  *[...]*

**3. Figure 2(a) shows that the two models are in good agreement, and they both have
some differences from GRACE. Does the input of precipitation significantly affect the
simulation results of the model? If other precipitation products are used as input, will
the results be different?**

**AC**: Thank you for raising an important point. In the original manuscript, we evaluated the
association of TWS variability with different precipitation products and found little influence
on the results (Fig. B8 and B9 in the manuscript).

Nevertheless, we have tested the robustness of the results of SINDBAD and H2M by
forcing the models using an independent precipitation estimate from MSWEP v.2.8  (Table 1
in the manuscript) instead of GPCP1dd, and repeated the whole analysis. We could again
verify that using another precipitation forcing does not significantly change the results and
findings of our study. We will highlight this finding in the manuscript and include the relevant
analyses (Fig. 1~5, below) in the appendix.

Specifically, two models are still in good agreement when they are forced by MSWEP. The
performance of two models has been slightly improved, shown by $R^2$, the distribution of
errors, and the slope of regression equations (Fig. 1). However, the main findings and
conclusions are not affected by that and stay the same. For example, spatial contribution to
TWS IAV (Fig. 2) and its modeling error (Fig. 3) remain similar in general, so do the
systematic larger $wRiver_{max}$ (Fig. 4) and more contribution of remotely-recharged
groundwater to transpiration (Fig. 5) in error hotspots.

This is in line with Kraft et al. (2022) who showed that H2M gave almost the same
performance when it is trained on different precipitation products.

[revised manuscript text omitted]

**4. The abscissa and ordinate of the scatter plot in Figure 3 have no text description**

**AC**: The text labels for the x and y axes were omitted for clarity and stated only in the figure caption. We further improved the related part of figure caption which now reads:

*[...] Below the diagonal, scatter plots comparing the pixel-wise contributions of the corresponding column (x-axis) versus row (y-axis) are shown. [...]*

**5. How much different precipitation inputs affect the modeling error of global terrestrial water storage interannual variability? Does the precipitation input or the different model structure affect the simulation error more?**

**AC**: As in the response to the comment 3, we agree that model input and structure are definitely important aspects that may potentially affect the results.

First, regarding the precipitation input, we show that the use of different precipitation products does not significantly alter the main results and findings of the study. This was shown in Kraft et al. (2022) as well.

Second, the potential effect of model structure is large. We assume that the two different models used in this study cover that source of uncertainty to a certain extent. Interestingly, we found that both models showed a large consistency in the findings despite having vastly different model structures with SINDBAD rooted on traditional hydrological concepts, and H2M formulated on modern machine learning methods.

Despite showing and presenting the effects of each of these factors separately, we cannot say with a large confidence if the uncertainty due to input is larger than that due to model structure or vice versa, especially based on what is presented in the current manuscript. For such an analysis, one would envisage a comprehensive factorial analysis of different modeling structures as well as the use of different input data but within a consistent seamless framework rather than comparison of two or more different models (as presented here, and in many model intercomparison projects to date). We will clarify this in the discussion with the following:

*Lastly, the additional sources of errors include 1) anthropogenic influence, 2) uncertainties in the GRACE and forcing data, and 3) model structure. In Africa, a decrease in TWS IAV in 2003–2006 (Fig. 6c) was due to the expansion of Nalubaale Dam as well as La Niña (Stager et al., 2007; Awange et al., 2013, 2019). [...]*

*With respect to the uncertainty from forcing and model structure, we show that the use of different precipitation products does not significantly alter the main results and findings of the study (Fig. B14-B18). This was shown in Kraft et al. (2022) as well. In addition, the potential effect of model structure is large. We assume that the two different models used in this study cover that source of uncertainty to a certain extent. Interestingly, we found that both models, which have vastly different model structures, showed a large consistency in the findings. Despite showing and presenting the effects of each of these factors separately, we cannot conclude with a large confidence if the uncertainty due to input is larger than that due to model structure or vice versa, especially based on what is presented in the current manuscript. For such an analysis, one would envisage a comprehensive factorial analysis of different modeling structures as well as the use of different input data but within a consistent seamless framework.*

, as well as in the introduction:

*Both modeling frameworks are heavily rooted on using observations, and include GRACE observations in the model parameter estimation and evaluation. As such, under ideal conditions, the models have a potential to simulate aspects of hydrological cycle that agree the most with relevant observations, and the model errors, if any, could be attributed to either model structure (e.g., missing model processes or the way to formulate and connect processes) or observational uncertainties.*

**Author's response to Referee #2**

**The authors presented a study to diagnose the modeling errors by comparing GRACE and model TWSA based on IAV. The motivation of this study is nice, since Scanlon's PNAS study revealed an interesting question on the discrepancy between GRACE and models. The focus on interannual is a good complementary to the focus on trend by Scanlon et al.. Generally, this study is interesting. However, I have some critical questions related to the methods used for analysis, which may largely affect the reliability of the findings.**

**AC**: We would like to thank the reviewer for a positive outlook of the study. Please find our responses below, which will also be included in the revised manuscript.

- **Generally, WGHM, PCR-GLOBWB, and maybe some other LSMs that include GW module, are more popularly used than the two models used in this study. I am not going to say the two models used here is not good enough, but I guess many researchers would be more interested on what will it like if we use WGHM, or PCR-GLOBWB, or CLSM. Besides, it is not clear how the including of GRACE in model parameter estimation and evaluation (Line 77) will impact the comparison between GRACE and the two models.**

  **AC:** While we agree that it will be interesting to look at the results of sophisticated land surface models or hydrological models, we focused on the two models because they are data-driven and represent the best-case scenario in terms of model performance against state-of-the-art observations. Note that Kraft et al. (2022) and Trautmann et al. (2022) conducted a direct comparison of SINDBAD and H2M, respectively, with global hydrological models from the eartH2Oserve ensemble (Schellekens et al., 2017), which reveal that the two models are at least in par or better than the GHMs in the Earth2O ensemble. So, the analysis presented in our study can be expected to be representative of other GHMs as well.

  Instead, the aim in our study was to go beyond the good performances of the two models used, and rather understand if there are underlying model assumptions and shortcomings that result in error of interannual variability of the global TWS. For this, we also needed the models to be forced by identical data, and be given a fair opportunity to learn from the observation. In fact, use of GRACE data in parameter estimation, theoretically, allows for the modeling framework to produce TWS simulations with no error. Note that this would not be possible with other GHMs/LSMs with uncalibrated parameters. We included this in the introduction and methods sections of the original manuscript and will improve the introduction in the revision as:

  *Both modeling frameworks are heavily rooted on using observations, and include GRACE observations in the model parameter estimation and evaluation. As such, under ideal conditions, the models provide the simulations that agree the most with observations, and the model errors, if any, could be attributed to either missing model*

*processes or observational uncertainties. As we employ a covariance matrix analysis (see Sect. 2.1.2), we not only evaluate the global IAV, but also identify the regions that are most relevant to the IAV of global TWS in GRACE observations and the two models. SINDBAD and H2M are appropriate for this purpose as it requires models to be forced by identical data, and be given a fair opportunity to learn from the observation. In addition, as two models cover different aspects of modeling approach (i.e., process-based vs. physics-guided machine learning), using SINDBAD and H2M can cover the uncertainty of model structure to a certain extent.*

Lastly, we, in fact, think that the framework presented in our study can be used for every model in the earth2Observe ensemble to identify the regions of largest TWS interannual variability and its error, which can then be utilized for model improvements. But, this is out of scope of the current study. Instead, we analyzed the error hotspots of TWS IAV modeling by four GHMs in the earth2Observe ensemble (W3RA, LISFLOOD, SURFEX-TRIP, and PCR-GLOBWB) as shown in Fig. 1 below. Four models show strong positive contributions in the humid regions of northern South America as SINDBAD and H2M do, but the four models disagree with SINDBAD and H2M for some aspects like hotspots in Central Africa. However, a direct comparison of the results of SINDBAD and H2M with those of the four GHMs are not reasonable because different products of forcing were used, and observations were either not used at all, or only used for model validation. We include Figure 1 as appendix, and we will address the issue in the discussion as:

*[...] In SINDBAD, vegetation indirectly accesses secondary water storage with capillary rise, which contributes to a larger evapotranspiration over some regions, including the regions in Africa (Fig. 9 in Trautmann et al., 2022).*

*As shown above, the model structure is an influential uncertainty of the location of hotspots. Though using SINDBAD and H2M cover the uncertainty to a certain extent, more sophisticated hydrological models may have different hotspots and sources of errors. Following Kraft et al. (2022), among 10 global hydrological models in the eartH2Observe ensemble, we selected four GHMs with groundwater storage in the structure to identify the hotspots of TWS IAV modeling error: W3RA (Van Dijk and Warren, 2010), LISFLOOD (Van Der Knijff et al., 2010), SURFEX-TRIP (Decharme et al., 2010, 2013), and PCR-GLOBWB (Van Beek et al., 2011; Wada et al., 2014). In Fig. B19, the four GHMs show strong positive contributions in the humid regions of northern South America as SINDBAD and H2M do, but the four models disagree with SINDBAD and H2M for some regions like hotspots in Central Africa. However, as different sets of forcing and constraints with different spatiotemporal domains were used for the simulation of the four GHMs, and given the complexity of their structure, further research will be required to investigate the hotspots and sources of TWS IAV modeling errors of each GHM.*

[Figure]

Figure 1. **The same as Fig. 4 in the manuscript, but for four global hydrological models in the eartH2Observe ensemble.** Global distribution of pixel-wise contributions to the variance of the modeling error of global terrestrial water storage interannual variability. Along the diagonal, maps of the pixel-wise contribution to the global TWS IAV modeling errors in W3RA, LISFLOOD, SURFEX-TRIP, and PCR-GLOBWB are shown. Above the diagonal, a map of the difference (i.e., column - row) is shown.

- **It is not clear why using Equation (1) to derive the IAV for analysis. I cannot understand the physical meaning of subtracting long-term trend (fit ()) from monthly values. So, the question comes that what is interannual variability, and how to define it? Can we just subtracting long-term average from monthly values? I am not sure my understanding is correct or not. Please verify it.**

**AC:** Thank you for pointing this out. It is important to define a term clearly and evaluate the model at a proper aspect. First, we think that subtracting the long-term average from monthly values is not suitable because it cannot remove the trend. We want to remove the trend because SINDBAD and H2M do not properly account for the trend as it is significantly driven by human activities (Rodell et al., 2018; Scanlon et al., 2018) and long-term processes such as vegetation (Pokhrel et al., 2021) and glacier melt (Rodell et al., 2018; Scanlon et al., 2018). Instead, by interannual variability, we want to quantify how much each value deviates from the seasonal

mean condition including the trend as Fig. 2 illustrates below. This definition of IAV will also make this study more suitable as a complementary to the study by Scanlon et al. (2018) as well as other relevant studies (e.g., Jung et al., 2017; Humphrey et al., 2018). To this goal in mind, for each month of a year, we calculated the linear regression fit that represents the seasonal mean value including trend. We then got the IAV by subtracting the fitted value from each monthly TWS.

For clarification, we will add Fig. 2 as appendix and the definition of IAV above in the manuscript as follows:

*In this study, IAV quantifies how much a value (e.g., TWS) deviates from the seasonal mean including the trend. Accordingly, we calculated the globally integrated GRACE TWS IAV as follows: [...]*

[Figure]

Figure 2. Illustration of the calculation of interannual variability for the global terrestrial water storage (TWS) anomalies.

- **Since GRACE Level-3 data has been already processed by subtracting the mean of a period (2004-2009?) from monthly TWS to get TWSA. If the authors again do subtracting (2002-2017) for GRACE and models, it may lead to mismatch between GRACE and model, because different subtracting were done for GRACE (subtracting 2004-2009, and then subtracting 2002-2017) and models (subtracting 2002-2017).**

**AC:** The reviewer is correct that the time period used to calculate the TWS anomaly is crucial and different time periods would affect the comparison between GRACE data and models. Exactly due to this difference, a suggested necessary step in the GRACE data usage is to align both GRACE and modeled TWSA to the same time mean anomaly (https://grace.jpl.nasa.gov/about/faq/), which, in the study, is 2002-2017. By removing the time period 2002-2017 from each series, both time series of TWS anomalies are consistent and reflect the deviations from the same baseline condition.

- **Line 128: I am not sure it is the best way to evaluate model performance by comparing the IAV derived from GRACE and models. How about compare TWSA?**

**AC:** We agree that TWSA can be evaluated as well, but as clearly mentioned in the introduction, the main aim of the study is to diagnose the error in IAV of TWS, which is still reproduced relatively poorer in the models compared to TWSA. The evaluation of TWS IAV presented here complements previous studies evaluating the anomalies (e.g., Scanlon et al., 2018) with climatic processes, and trends with anthropogenic influences. We also note that the original TWSA includes the trend that should be removed before the model evaluation as SINDBAD and H2M do not account for important processes that affect the trend, e.g., human influences.

- **Before Figure 2, people may be interested on seeing spatial distribution map of TWSA from GRACE and models, as well as the distribution map of IAV, which both can help we better understand the difference and consistence between GRACE and models.**

**AC:** Thank you for the suggestion. We will add figures for spatial distribution of TWSA (Fig. 3, below) and TWS IAV (Fig. 4, below) to appendix, and will mention them in the manuscript as follows:

*SINDBAD and H2M reasonably reproduce the observed* time series of *global TWS IAV by GRACE ($R^2$ of 0.49 and 0.51 for SINDBAD and H2M, respectively) (Fig. 2),* as well as the spatial pattern of TWS anomaly and TWS IAV (Fig. B20 and B21). *[...]*

[Figure]

Figure 3. Global distribution of the standard deviation (std) of the global terrestrial water storage (TWS) anomalies. Along the diagonal, maps of the pixel-wise std of TWS anomalies in GRACE, SINDBAD, and H2M are shown (indicated by the label of row or column). Above the diagonal, maps of the difference (i.e., column - row) are shown. For example, the map of the first row and the second column is for SINDBAD (column) minus GRACE (row). Below the diagonal, scatter plots comparing the corresponding column (x-axis) versus row (y-axis) are shown. In the scatter plots, colors indicate the density of points, *r* is the Pearson correlation coefficient and *ρ* is the Spearman correlation coefficient. Red lines are linear regression fit and red texts are corresponding equations. White pixels within land boundaries in maps are invalid as they are out of the study area.

[Figure]

Figure 4. Global distribution of the standard deviation (std) of the global terrestrial water storage (TWS) interannual variability (IAV). Along the diagonal, maps of the pixel-wise std of the global TWS IAV in GRACE, SINDBAD, and H2M are shown (indicated by the label of row or column). Above the diagonal, maps of the difference (i.e., column - row) are shown. For example, the map of the first row and the second column is for SINDBAD (column) minus GRACE (row). Below the diagonal, scatter plots comparing the corresponding column (x-axis) versus row (y-axis) are shown. In the scatter plots, colors indicate the density of points, $r$ is the Pearson correlation coefficient and $\rho$ is the Spearman correlation coefficient. Red lines are linear regression fit and red texts are corresponding equations. White pixels within land boundaries in maps are invalid as they are out of the study area.

- **Figure 3: Sorry, but I do feel difficult to understand what the exact meanings of the spatial maps are. Maybe more information can be added to the figure showing who minus who, something like that. Besides, I guess the white blank areas here are the grid cells with positive covariances, is it true?**

**AC:** Thank you for pointing out the confusion. The detail (i.e., who minus who) was omitted for clarity, but was explained in the caption. We will improve the caption of Fig. 3 and 4 in the manuscript as follows:

*Figure 3. Global distribution of pixel-wise contributions to the variance of the global terrestrial water storage (TWS) IAV. Along the diagonal, maps of the pixel-wise contribution in GRACE, SINDBAD, and H2M are shown (indicated by the label of row or column). Above the diagonal, maps of the difference (i.e., column - row) are shown. For example, the map of the first row and the second column is for SINDBAD (column) minus GRACE (row). [...] Red lines are linear regression fit and red texts are corresponding equations. White pixels within land boundaries are invalid as they are out of the study area.*

*Figure 4. Global distribution of pixel-wise contributions to the variance of the modeling error of global terrestrial water storage interannual variability. Along the map of the first row and the second column is for SINDBAD (column) minus GRACE (row). [...] White pixels within land boundaries are invalid as they are out of the study area.*

We will also add the same sentence about white pixels to captions of relevant figures such as Fig. B3 and B12 in the manuscript and ones that will be added in the revised manuscript.

**Author's response to Referee #3**

This study models the error between the global TWS IAV observations of GRACE and two models, SINDBAD and H2M. The authors found that the global TWS IAV is mainly driven by humid tropical and semi-arid regions, and identified the hotspots of modeling errors of the global TWS IAV mainly in tropical regions that span across climatic regions. The study presents a novel way to attribute global variability to each pixel and focused on regions where hydrological cycle components in models may not be sufficiently well represented due to their complex hydrological and climatological processes.

The study in general is well-written and easy to follow.

**AC:** We would like to express our gratitude to the reviewer for positive feedback and suggestions on the manuscript. Of course, we have responded to the other reviewers and will incorporate all suggestions into the revised manuscript. Below, you will find the responses to each comment of reviewer 3.

Additional to comments made by the two Anonymous Referees, which I consider important to answer, my comments are as follows:

- **As the study identifies humid regions of northern South Americas as one of the main drivers of global TWS IAV, I suggest including these references in the discussion in which global models are compared with GRACE products in a very important instrumented tropical basin.**

Bolaños Chavarría, S., Werner, M., Salazar, J. F., & Betancur, T. (2022). Benchmarking global hydrological and land surface models against GRACE in a medium-sized tropical basin. Hydrology and Earth System Sciences, 26(16), 4323-4344.

Bolaños, S., Salazar, J. F., Betancur, T., & Werner, M. (2021). GRACE reveals depletion of water storage in northwestern South America between ENSO extremes. Journal of Hydrology, 596, 125687.

   **AC:** Thank you for pointing to the relevant studies. We will include them in the discussion of revised manuscript as:

*On the other hand, tropical regions come out as the dominant contributor to the variance of the global TWS IAV modeling errors (Fig. 4). Tropical regions were reported as one significant contributor to the global TWS IAV, but with a large disparity between the models and GRACE (Humphrey et al., 2018) due possibly to characteristics of the regions that the tested models do not properly account for, for example, artificial reservoirs, complex topography, and wetlands (Bolaños et al., 2021, 2022). [...]*

- **I am a bit confused with Equation 1, in figure 1 I think it is clear that TWS IAV is the result of detrending and deseasonalizing TWS, but in Equation 1, I understand that only TWS is deseasonalized.**

    **AC:** Yes, as in Fig. 1 in the manuscript, Eq. (1) deseasonalizes and detrends TWS as the linear fitting includes the trend of the month of a year across years. Eq. (1) could detrend as well because each regression line of a month includes the trend as Fig. 1 illustrates below. We will include Fig. 1 in the revised manuscript as appendix.

[Figure]

Figure 1. Illustration of the calculation of interannual variability for the global terrestrial water storage (TWS) anomalies.

- **I think is necessary to define what is the meaning of SREX Regions, I don't identify what is.**

**AC:** We will add the meaning of SREX in the manuscript as follows:

*After the error hotspots are identified, we compare the time series of TWS and precipitation IAVs at the regional scale for error hotspots within selected Intergovernmental Panel on Climate Change (IPCC) Special Report on Extremes (SREX) regions (Sect. 3.4; see Fig. B1 for the SREX regions) to diagnose TWS IAV errors. [...]. Note that SREX regions include different regions of the world, and they have been used extensively to diagnose regional variation of climate model simulations (e.g., Seneviratne et al., 2012; Pokhrel et al., 2021).*

and in the caption of Fig. B2 of the manuscript as follows:

*Figure B1. The Intergovernmental Panel on Climate Change (IPCC) Special Report on Extremes (SREX) regions (Seneviratne et al., 2012), which were used in figures (Figs. 6, B4, B6, B7, and B8) to spatially average global terrestrial water storage interannual variability time series into four selected regions: the Laurentian Great Lakes (SREX regions 5), Amazon (SREX regions 7), Eastern and Western Africa (SREX regions 15 and 16), and South Asia (SREX regions 23).*

- **Why the preference for the JPL mascon if there is another mascon product like the mascon CSR that has the same resolution?**

**AC:** The purpose of this study is to qualitatively diagnose the hotspots of the global TWS IAV and its modeling error. For this purpose, either JPL mass concentration (mascon) product or CSR mascon product can be used, as JPL mascon and CSR mascon are qualitatively comparable to each other across global basins and at the interannual scale as well as longer-term temporal scales (Scanlon et al., 2016).

Nevertheless, to further clarify this, we analyzed the spatial contribution to the global TWS IAV for RL06 version 1 mascon GRACE products by JPL (i.e., one used in this study) and CSR (i.e., a comparable one by CSR). We find that the two GRACE products are largely consistent with each other in terms of hotspots of the dynamics of global TWS IAV (Fig. 2). This suggests that the main findings of our study would not change even if a different GRACE data product was used.

Scanlon, B. R., Zhang, Z., Save, H., Wiese, D. N., Landerer, F. W., Long, D., Longuevergne, L., and Chen, J. (2016), Global evaluation of new GRACE mascon products for hydrologic applications, *Water Resour. Res.*, 52, 9412– 9429, doi:10.1002/2016WR019494.Note that SREX regions include different regions

[Figure]

Figure 2. **The same as Fig. 3 in the manuscript, but using two mascon GRACE products only.** Global distribution of pixel-wise contributions to the variance of the global terrestrial water storage (TWS) IAV. Along the diagonal, maps of the pixel-wise contribution in GRACE by JPL and GRACE by CSR are shown. Above the diagonal, maps of the difference (i.e., column - row) are shown. Below the diagonal, scatter plots comparing the corresponding column (x-axis) versus row (y-axis) are shown. In the scatter plots, colors indicate the density of points, $r$ is the Pearson correlation coefficient, and $\rho$ is the Spearman correlation coefficient. Red lines are linear regression fit and red texts are corresponding equations.

**Save, H., S. Bettadpur, and B.D. Tapley (2016), High resolution CSR GRACE RL05 mascons, J. Geophys. Res. Solid Earth, 121, doi:10.1002/2016JB013007.**

- **Figure 2 a) describes a "NSE is the Nash-Sutcliffe Efficiency", but it does not appear in the figure**

**AC:** We are sorry for the error. We will remove the wrong reference from the caption as follows:

*Figure 2. [...] $R^2$ statistics in the bottomleft is calculated as the square of the Pearson correlation coefficient; NSE is the Nash-Sutcliffe Efficiency. [...].*

We will correct other errors as well, for example, the second text of Fig. 2a, from $R^2$(GRACE, SINDBAD) to $R^2$(GRACE, H2M).

---

## Editor Decision (ED1)

Hydrol. Earth Syst. Sci. Discuss., referee comment RC1
https://doi.org/10.5194/hess-2022-284-RC1, 2022
**Comment on hess-2022-284**

Anonymous Referee #1

Referee comment on "Diagnosing modeling errors of global terrestrial water storage interannual variability" by Hoontaek Lee et al., Hydrol. Earth Syst. Sci. Discuss., https://doi.org/10.5194/hess-2022-284-RC1, 2022

This study quantified the contribution of each pixel to the global TWS IAV of GRACE observations and two selected predominantly data-driven models, SINDBAD and H2M, as well as its modeling errors. The results show that the global TWS IAV is mainly driven by humid tropical and semi-arid region. The hotspots of modeling errors of the global TWS IAV are mainly located in tropical regions that span across climatic regions. The study provides an improved understanding of the global TWS IAV and its modeling error. Generally, the topic is important, and the study is well written and easy to follow. My comments are as follows.

1. In the high latitudes of the northern hemisphere, glacier changes contribute to TWS, whether the SINDBAD model and the H2M model have a glacier module.

2. It needs to be further pointed out that the model is inconsistent with GRACE in typical irrigation areas, such as the western United States, northern India, etc.

3. Figure 2(a) shows that the two models are in good agreement, and they both have some differences from GRACE. Does the input of precipitation significantly affect the simulation results of the model? If other precipitation products are used as input, will the results be different?

4. The abscissa and ordinate of the scatter plot in Figure 3 have no text description

5. How much different precipitation inputs affect the modeling error of global terrestrial water storage interannual variability? Does the precipitation input or the different model structure affect the simulation error more?

[Figure]

Hydrol. Earth Syst. Sci. Discuss., referee comment RC2
https://doi.org/10.5194/hess-2022-284-RC2, 2022
**Comment on hess-2022-284**

Anonymous Referee #2
* * *
Referee comment on "Diagnosing modeling errors of global terrestrial water storage interannual variability" by Hoontaek Lee et al., Hydrol. Earth Syst. Sci. Discuss., https://doi.org/10.5194/hess-2022-284-RC2, 2022
* * *
The authors presented a study to diagnose the modeling errors by comparing GRACE and model TWSA based on IAV. The motivation of this study is nice, since Scanlon's PNAS study revealed an interesting question on the discrepancy between GRACE and models. The focus on interannual is a good complementary to the focus on trend by Scanlon et al.. Generally, this study is interesting. However, I have some critical questions related to the methods used for analysis, which may largely affect the reliability of the findings.

- Generally, WGHM, PCR-GLOBWB, and maybe some other LSMs that include GW module, are more popularly used than the two models used in this study. I am not going to say the two models used here is not good enough, but I guess many researchers would be more interested on what will it like if we use WGHM, or PCR-GLOBWB, or CLSM. Besides, it is not clear how the including of GRACE in model parameter estimation and evaluation (Line 77) will impact the comparison between GRACE and the two models.
- It is not clear why using Equation (1) to derive the IAV for analysis. I cannot understand the physical meaning of subtracting long-term trend (fit ()) from monthly values. So, the question comes that what is interannual variability, and how to define it? Can we just subtracting long-term average from monthly values? I am not sure my understanding is correct or not. Please verify it.
- Since GRACE Level-3 data has been already processed by subtracting the mean of a period (2004-2009?) from monthly TWS to get TWSA. If the authors again do subtracting (2002-2017) for GRACE and models, it may lead to mismatch between GRACE and model, because different subtracting were done for GRACE (subtracting 2004-2009, and then subtracting 2002-2017) and models (subtracting 2002-2017).
- Line 128: I am not sure it is the best way to evaluate model performance by comparing the IAV derived from GRACE and models. How about compare TWSA?
- Before Figure 2, people may be interested on seeing spatial distribution map of TWSA from GRACE and models, as well as the distribution map of IAV, which both can help we better understand the difference and consistence between GRACE and models.
- Figure 3: Sorry, but I do feel difficult to understand what the exact meanings of the spatial maps are. Maybe more information can be added to the figure showing who minus who, something like that. Besides, I guess the white blank areas here are the

grid cells with positive covariances, is it true?

[Figure]

Hydrol. Earth Syst. Sci. Discuss., referee comment RC3
https://doi.org/10.5194/hess-2022-284-RC3, 2022
**Comment on hess-2022-284**

Anonymous Referee #3
* * *
Referee comment on "Diagnosing modeling errors of global terrestrial water storage interannual variability" by Hoontaek Lee et al., Hydrol. Earth Syst. Sci. Discuss., https://doi.org/10.5194/hess-2022-284-RC3, 2022
* * *
This study models the error between the global TWS IAV observations of GRACE and two models, SINDBAD and H2M. The authors found that the global TWS IAV is mainly driven by humid tropical and semi-arid regions, and identified the hotspots of modeling errors of the global TWS IAV mainly in tropical regions that span across climatic regions. The study presents a novel way to attribute global variability to each pixel and focused on regions where hydrological cycle components in models may not be sufficiently well represented due to their complex hydrological and climatological processes.

The study in general is well-written and easy to follow. Additional to comments made by the two Anonymous Referees, which I consider important to answer, my comments are as follows:

- As the study identifies humid regions of northern South Americas as one of the main drivers of global TWS IAV, I suggest including these references in the discussion in which global models are compared with GRACE products in a very important instrumented tropical basin.

Bolaños Chavarría, S., Werner, M., Salazar, J. F., & Betancur, T. (2022). Benchmarking global hydrological and land surface models against GRACE in a medium-sized tropical basin. *Hydrology and Earth System Sciences*, *26*(16), 4323-4344.

Bolaños, S., Salazar, J. F., Betancur, T., & Werner, M. (2021). GRACE reveals depletion of water storage in northwestern South America between ENSO extremes. *Journal of Hydrology*, *596*, 125687.

- I am a bit confused with Equation 1, in figure 1 I think it is clear that TWS IAV is the result of detrending and deseasonalizing TWS, but in Equation 1, I understand that only TWS is deseasonalized.
- I think is necessary to define what is the meaning of SREX Regions, I don't identify what is.
- Why the preference for the JPL mascon if there is another mascon product like the mascon CSR that has the same resolution?

Save, H., S. Bettadpur, and B.D. Tapley (2016), High resolution CSR GRACE RL05 mascons, J. Geophys. Res. Solid Earth, 121, doi:10.1002/2016JB013007.

- Figure 2 a) describes a "NSE is the Nash-Sutcliffe Efficiency", but it does not appear in the figure